# Committee machines—a universal method to deal with non-idealities in memristor-based neural networks

D. Joksas [1✉], P. Freitas[2], Z. Chai[2], W. H. Ng [1], M. Buckwell[1], C. Li[3], W. D. Zhang [2], Q. Xia [3], A. J. Kenyon[1] & A. Mehonic [1✉]

Artificial neural networks are notoriously power- and time-consuming when implemented on conventional von Neumann computing systems. Consequently, recent years have seen an emergence of research in machine learning hardware that strives to bring memory and computing closer together. A popular approach is to realise artificial neural networks in hardware by implementing their synaptic weights using memristive devices. However, various device- and system-level non-idealities usually prevent these physical implementations from achieving high inference accuracy. We suggest applying a well-known concept in computer science—committee machines—in the context of memristor-based neural networks. Using simulations and experimental data from three different types of memristive devices, we show that committee machines employing ensemble averaging can successfully increase inference accuracy in physically implemented neural networks that suffer from faulty devices, device-to-device variability, random telegraph noise and line resistance. Importantly, we demonstrate that the accuracy can be improved even without increasing the total number of memristors.

[1] Department of Electronic and Electrical Engineering, University College London, Roberts Building, Torrington Place, London WC1E 7JE, UK. [2] Department of Electronics and Electrical Engineering, Liverpool John Moores University, Liverpool, James Parsons Building, Byrom Street, Liverpool L3 3AF, UK. [3] Department of Electrical and Computer Engineering, University of Massachusetts Amherst, 100 Natural Resources Road, Amherst, MA 01003-9292, USA. ✉email: dovydas.joksas.15@ucl.ac.uk; adnan.mehonic.09@ucl.ac.uk

Artificial neural networks (ANNs), with all of their variants, are now the main tools in machine learning tasks, such as classification. The vast amounts of data being constantly produced have enabled successful training and operation of ANNs. However, to achieve high inference accuracy, it is usually necessary for neural networks to have a large number of parameters. This results in both training[1] and inference[2] stages being time and power consuming. This is largely caused by the need to transfer data from memory to computing units—physical separation of memory and computing is the essence of any von Neumann system.

One of the most promising solutions to these problems is the paradigm of non-von Neumann computing and, specifically, analogue implementations of synapses (weights) in physical ANNs. Because there are many more synapses than there are neurons in ANNs, the matrix-vector multiplications, in which the synaptic weight values are used, are the costliest operations in these networks, both in terms of power and time. Computing directly in memory would minimise data transfers from off-chip memory, thus the most popular approach is using analogue memory devices as proxies for synaptic weights of ANNs (both fully connected and their variants[3,4]). A common technique is to arrange such devices in a structure, called crossbar array, in which every device (or a pair of devices) is used to represent a single synaptic weight or, more generally, an entry in a matrix[5]. Memristive devices, such as phase-change memories (PCMs)[6,7] or resistive random-access memories (RRAMs)[8,9], have been considered as candidates for such tasks. Although here we focus on ex situ training, such systems have been successfully utilised for in situ training too[10,11].

In memristive implementations of ANNs, the main concern is that various non-idealities associated with these devices can prevent these systems from achieving high accuracy[12,13]. Examples of non-idealities affecting inference accuracy include, but are not limited to, devices not being able to electroform, devices stuck in one of the resistance states after electroforming, device-to-device (D2D) variability and random telegraph noise (RTN). When training analogue systems in situ, limited endurance and nonlinear resistance modulation too have to be taken into account. To mitigate the effects of these device non-idealities, it is often necessary to modify device structure[9], to use more advanced programming schemes[14] or to use additional circuitry[15] or high-precision processing units[16] in conjunction with memristive elements. On the system level, there is an issue of line resistance which affects the distribution of currents and thus decreases the accuracy. These line resistance effects can be partially compensated for algorithmically[17] or partially mitigated by using multiple smaller crossbar arrays[18]. Examples of past efforts at dealing with these and other non-idealities of memristive devices and systems are listed in Table 1; most of these non-idealities are still the main focus of the research in the neuromorphic community.

We propose a simple way to mitigate the effects of all types of non-idealities during inference. We suggest combining several non-ideal memristor-based neural networks into committees to achieve better accuracy. The committee machine (CM) method we propose significantly increases the inference accuracy and does not increase the computation time because memristive ANNs in such committees work in parallel.

In this work, we firstly explain the simulation setup—what networks were trained, how they were simulated and how they were combined into CMs. After that, follows the experimental part. We investigate three different types of memristor technology —tantalum/hafnium oxide-based (Ta/HfO$_2$), tantalum oxide-based (Ta$_2$O$_5$), and amorphous vacancy modulated conductive oxide-based (aVMCO) devices. By exploring their non-idealities relevant to inference—faulty devices, D2D variability, RTN, and

line resistance—we use the experimental data to simulate memristive ANNs working individually and in committees.

## Results

**Simulation setup.** Fully connected ANNs were trained in software to recognise handwritten digits (using MNIST data base[19]). Architectures with one hidden layer were explored. Unless stated otherwise, the simulations used networks with 25 hidden neurons. However, networks with 50, 100, and 200 hidden neurons were additionally employed to evaluate the effectiveness of the proposed method while controlling for the total number of memristors required. Following training, weights of ANNs were mapped onto pairs of conductances using proportional mapping scheme (see ref. [20]) to simulate memristor-based ANNs. Finally, these memristive networks were disturbed using experimental data to reflect the effect of device- and system-level non-idealities.

After simulating physical non-idealities, the networks were combined into CMs that employed ensemble averaging (EA)[21]. The principle of EA is shown in Fig. 1a—several networks are combined in parallel and then their outputs are averaged. After that, the prediction is made using the averaged vector—the prediction is the label corresponding to the largest entry in the vector.

CM methods are frequently used even with conventional ANNs. Methods, such as EA, often produce better accuracy than that of the best individual network in a committee[22]. Although there are other types of CMs besides EA, they often rely on training additional gating networks or boosting networks during the training stage. Using a gating network in this scenario would produce additional problems—to avoid it acting as a performance bottleneck, it too would have to be implemented on crossbar arrays. Various non-idealities would decrease the effectiveness of this gating network, which is responsible for making the decisions about the whole committee of ANNs. Likewise, we speculate that boosting of networks would not be feasible in ex situ training because it requires information about where individual ANNs perform poorly—this cannot be known precisely until they are implemented physically on crossbar arrays and the non-idealities manifest themselves. To authors' best knowledge, the application of boosting in the context of memristive neural networks seems to have been explored only once before[23]; as expected, it requires training each memristive implementation differently because non-idealities manifest themselves differently in different crossbar arrays.

There exist modifications of EA algorithm that could potentially perform better. One example of this is generalized ensemble method (GEM) which, instead of using equal weightings for each network during averaging (as in EA), uses a different one for each network[21]. These weightings are analytically determined by considering correlation of errors between different networks. But because ref. [21] only considered networks with mean square error loss function (while our networks used cross-entropy loss function), this work does not explore GEM. Instead, we investigated whether it is possible to achieve a better performance by optimising the weightings numerically. This method, like GEM and others previously mentioned, might be impractical because, firstly, these weightings could be determined only after the ANNs are physically implemented on crossbars, and, secondly, the devices could change throughout their lifetimes thus affecting the optimal weightings.

Even with the assumption that the devices would have perfect retention, we found that optimisation of weightings achieves effectively the same performance. Because of these reasons, we focus only on EA in the main text, but present our results of optimising weightings in Supplementary Fig. 5. We stress that we

**Table 1 Examples of past efforts at dealing with non-idealities of memristive devices and their systems.**

| First author (year) | Non-ideality | Device type | Proposed solution |
|---|---|---|---|
| C. Sung (2018)[31] | Current/voltage nonlinearity | TaO$_x$ RRAM | Hot-forming step is adopted |
| C. Li (2018)[15] | Current/voltage nonlinearity | Ta/HfO$_2$ RRAM | 1T1R architecture is adopted |
| Y. Fang (2018)[32] | Device-to-device variability | HfO$_x$ RRAM | Ultra-thin ALD-TiN buffer layer is introduced |
| B. Govoreanu (2013)[33] | Device-to-device variability | Al$_2$O$_3$/TiO$_2$ (VMCO) RRAM | Non-filamentary RRAM is adopted |
| A. J. Kenyon (2019)[34] | Device-to-device variability | SiO$_x$ RRAM | The roughness of bottom electrodes is increased |
| L. Xia (2017)[14] | Faulty devices | – | A modified mapping algorithm and redundancy schemes are used |
| S. Ambrogio (2018)[7] | Limited dynamic range | PCM | Two pairs of conductance of varying significance for every synaptic weight are used |
| M. Hu (2016)[17] | Line resistance | – | Advanced mapping algorithms are used to compensate for line resistance effects |
| W. Wu (2018)[35] | Programming nonlinearity | HfO$_x$ RRAM | Electro-thermal modulation layer is deposited on the switching layer |
| J. Woo (2016)[9] | Programming nonlinearity | HfO$_2$ RRAM | Bilayer structure is adopted |
| S. Ambrogio (2018)[7] | Programming nonlinearity | PCM | PCM devices are used together with CMOS transistors |
| Z. Chai (2018)[36] | Random telegraph noise | TiO$_2$/a-Si (aVMCO) RRAM | Non-filamentary RRAM is adopted |

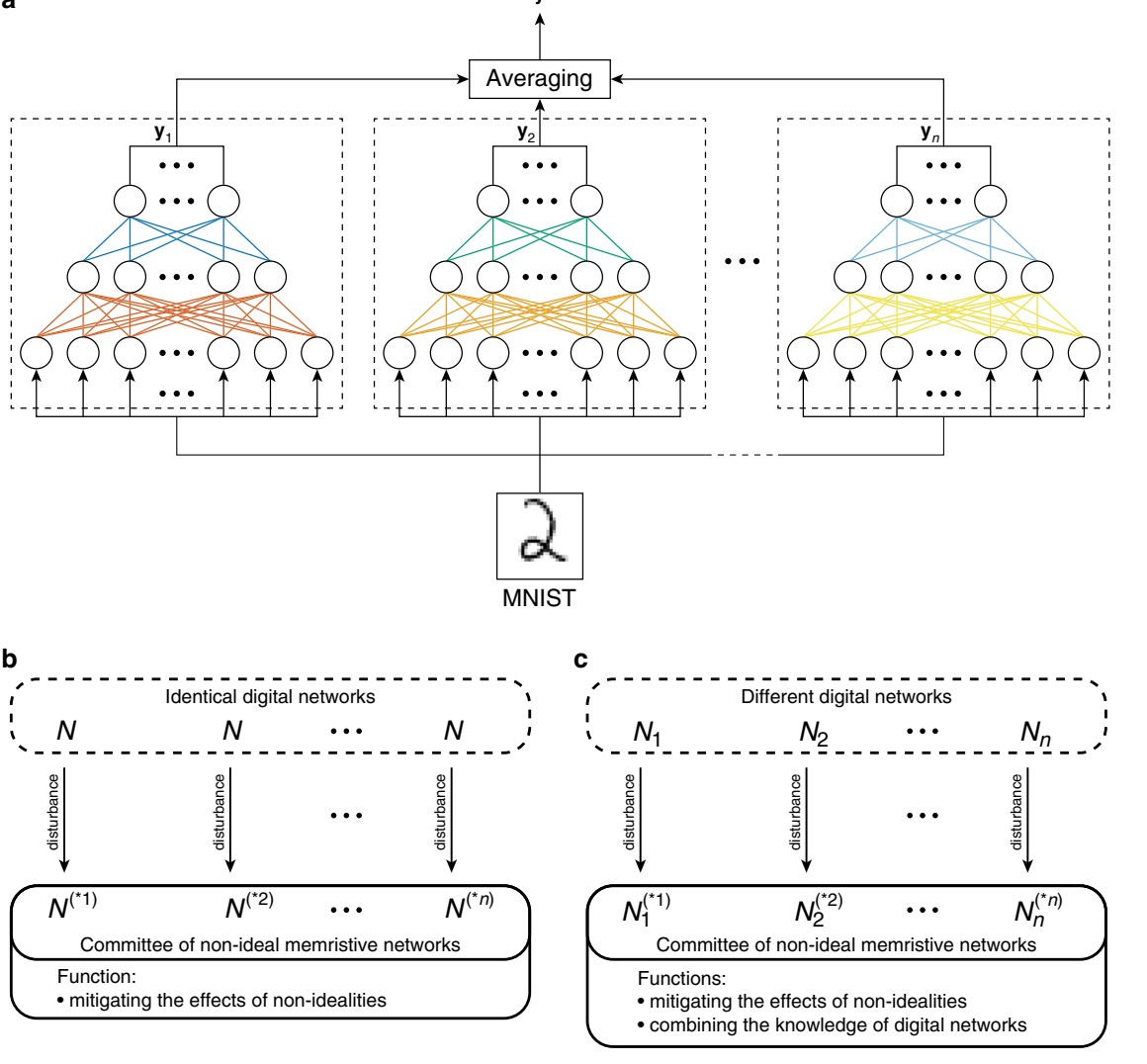

**Fig. 1 Using multiple neural networks to improve inference accuracy. a** The principle of ensemble averaging. **b** Using identical digital networks when implementing committees of memristive neural networks only helps to deal with the damage to the networks caused by the non-idealities. **c** Using different digital networks when implementing committees of memristive neural networks both helps to deal with the damage to the networks caused by the non-idealities and allows to combine the knowledge about the dataset acquired by individual digital networks.

are open to the idea that other CM methods besides EA could be utilised successfully for ex situ training in the context of memristive ANNs. However, in this work we focus on demonstrating that CMs can be used to improve the accuracy of memristor-based ANNs in general.

With EA, we find that even when the memristive ANNs, which go into a committee, all use the same digital weights that are mapped onto crossbar arrays (see Fig. 1b), committee of memristor-based networks can still achieve higher accuracy than just a single non-ideal network. Although all networks have the same digital weights before mapping, their physical implementations (which we call "disturbances" in Fig. 1b, c because they can usually be represented by the modification of individual weights) will be different. For example, in one crossbar array, a certain set of devices will be faulty, while in the other crossbar array, it will be a different set. This will result in different physical implementations having slightly different learned representations of the dataset, or, to paraphrase, different networks will be "damaged" differently by the non-idealities. This means that these committees will be able to combine different representations, and thus achieve higher accuracy. However, by definition, such approach would almost certainly not yield a committee accuracy that is higher than the accuracy of a single digitally implemented network.

A better approach is to use different digital networks for different physical implementations that go into a committee (see Fig. 1c). This approach much more resembles the conventional application of EA in computer science. In the context of memristive crossbar arrays, it would not only help to mitigate the effects of the non-idealities (as in the case of Fig. 1b), but would also allow to combine the representations of digital networks that were different even before the mapping stage. Most importantly, this method allows for a committee to achieve higher accuracy, which is sometimes even higher than that of individual networks with digitally implemented weights. We thus used this method in this analysis. An example comparison of these two approaches is presented in Supplementary Fig. 8.

In this work, any given committee used only one network architecture but each network was initialised differently before training, thus trained networks had different sets of weights. Although it was not explored in this work, combining different network architectures in a committee of memristor-based networks might be advantageous. Furthermore, in this work we focus on fully connected ANNs but CMs could be applied to other variants of neural networks as well. Due to the simplicity of EA, it could, for example, be employed in convolutional neural networks (CNNs)[24], which are often used for image classification. This might be of interest as CNNs have been successfully implemented using crossbar arrays recently[25]. However, crossbar implementations are naturally more suited to fully connected networks, therefore we limit ourselves to this architecture but are open to exploring the effectiveness of EA with memristive CNNs in the future.

**Ta/HfO$_2$ RRAM**. With array-level data available, Ta/HfO$_2$ experiments provide the most complete picture of device- and system-level non-idealities. In this subsection, we present not only the analysis of faulty devices and D2D variability, but also careful consideration of the line resistance effects. Ta/HfO$_2$ memristors do not exhibit apparent RTN and overall have excellent retention properties[26], and thus are perfect candidates for inference application.

Faulty devices and device-to-device variability: The most energy-efficient procedure to modulate the conductance of memristors is by the application of voltage pulses. In an ideal

scenario, one would apply identical pulses and observe constant increases in conductance with each pulse. This is rarely the case in practise but, fortunately, this type of behaviour is more relevant for in situ training where it is necessary to ensure linear adjustment of ANN's weights[27]. In ex situ training, conductance verification schemes can be used to program the devices precisely. Because the devices would have to be programmed only once, one can spend additional resources to do so accurately by applying SET (potentiation) and RESET (depression) pulses until a desirable conductance state is achieved.

Even with this approach, there remain two obstacles—faulty devices and D2D variability. It is observed in most memristor technologies that at least a small fraction of the devices tends to get stuck in a particular conductance state. Additionally, even if not stuck, different devices might behave differently; for example, they might have different conductance ranges. Figure 2a shows conductance changes in Ta/HfO$_2$ RRAM devices (in a $128 \times 64$ crossbar array) when they are applied with voltage pulses. We can see from the median values that overall the devices' conductance tends to increase as more SET pulses are applied. However, the wider bottom regions of the violin plots indicate that some devices are stuck around high resistance state (HRS) and cannot set entirely no matter how many voltage pulses are applied. There also exist devices that are stuck in low resistance state (LRS), or simply do not span the full conductance range.

Figure 2a combines data from multiple SET cycles for each of the memristors, thus it is important to understand how do these devices behave individually. Figure 2b–f show conductance of five (out of 8192) devices over 11 SET and RESET cycles. In the five diagrams, the radial component represents the conductance (in mS) and the angular component represents the number of applied pulses. Figure 2b shows an example of preferable (and typical) device behaviour—conductance changes in a continuous fashion and spans a wide range of conductance values, from ~0.1 mS to ~1.0 mS. Although RESET cycles tend to feature abrupt decreases in conductance, one can always repeat a cycle and exploit the more predictable behaviour of SET cycles.

When encoding continuous numbers into crossbar devices' conductances, it is often preferable to choose a large enough conductance range. Using data from Fig. 2a, one could, for example, choose the range between the first and the last median points (from ~0.1 mS to ~1.0 mS). Device, whose behaviour is presented in Fig. 2b, could be easily set to any conductance within that range, as we have seen before. On the other hand, device, whose behaviour is presented in Fig. 2c, although operating in a predictable fashion, has smaller conductance range. We can see that in all cycles, its conductance does not exceed 0.8 mS. This is an example of D2D variability that can make it difficult to choose optimal operating range and set the conductance of all devices precisely.

Device, whose behaviour is presented in Fig. 2d, shows high cycle-to-cycle variability. Although that could prove to be a problem in some applications, this specific device might perfectly serve its purpose in ex situ training of ANNs. We can observe that this device spans the same conductance range as device from Fig. 2b, even if in an unpredictable manner. Because all states in the full range are, in theory, achievable, one can cycle the device multiple times until it is set to the required conductance level.

Lastly, we have devices whose negative effect is most difficult to mitigate—faulty devices. Figure 2e shows behaviour of a device stuck at high conductance values, while Fig. 2f shows behaviour of a device stuck at low conductance values. No matter how many pulses the devices are applied with or how many times they are cycled, they exhibit almost no conductance variation and thus, in most cases, cannot be used to encode information.

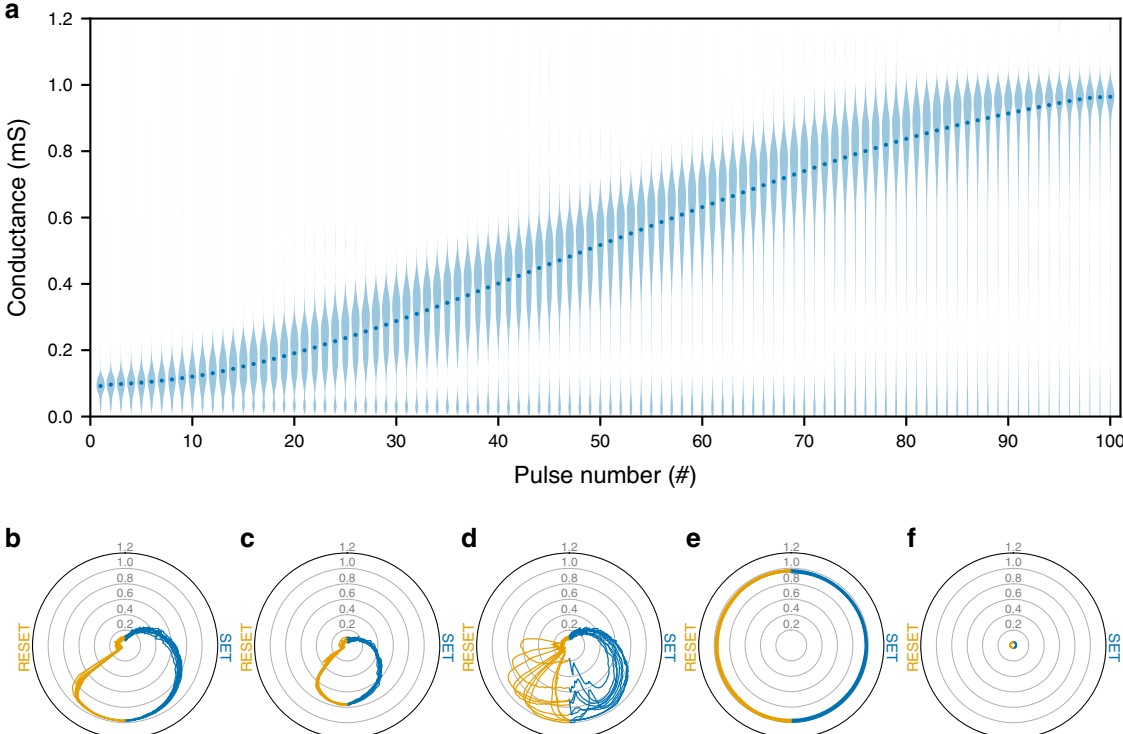

**Fig. 2 Experimental data of Ta/HfO$_2$ RRAM crossbar array.** Data of a crossbar of shape 128 × 64 were used. **a** Modulation of devices' conductance over 11 SET cycles, each consisting of a 100 potentiating pulses. Violin plots of gradual conductance changes are shown for all Ta/HfO$_2$ devices, with dots representing median conductance after a certain number of pulses. 100 points were used for Gaussian kernel density estimation. All violin plots have their maximum widths normalised. **b**–**f** Examples of devices with their conductance (in mS) **b** spanning the full range, **c** spanning part of the full range, **d** exhibiting cycle-to-cycle variability, **e** stuck at high values, and **f** stuck at low values. These diagrams show conductance of five devices from Ta/HfO$_2$ crossbar array over 11 SET and RESET cycles. The radial component represents the conductance, while the angular component represents the number of applied pulses. The first SET cycle starts at the top of each of the diagrams. The conductance (in blue) over 100 SET pulses is displayed in a clockwise fashion across the right half of each of the diagrams. Following that, conductance (in orange) over 100 RESET pulses (starting at the bottom) is displayed across the left half of each of the diagrams, after which the next cycle is displayed. Cartesian version of these plots is shown in Supplementary Fig. 9.

Knowing that some devices perform like the ones whose behaviour is shown in Fig. 2c, e, f, it is important to minimise their negative effect. If the conductance that a device has to be set to is outside that device's range, it is sensible to set it to the closest achievable conductance. Although there is little that can be done about fully stuck memristors, it is possible to optimise the behaviour of devices like the one in Fig. 2c that simply have smaller conductance range. For example, if such a device has to be set to 0.9 mS, one would set it to the highest achievable conductance (~0.8 mS). In the following simulations involving faulty devices and D2D variability, operating range between the first and the last median points was used, the devices were chosen randomly from the 128 × 64 crossbar and set to the most desirable states, as described in this paragraph.

Line resistance: The effect of line resistance can be extremely detrimental in many crossbar-based implementations of ANNs. That is especially the case if the crossbars used are large and the resistance of the interconnects is high (compared to memristors' resistance). Because in a neural network many of the inputs are non-zero at any given time, a lot of current accumulates in the bit lines, which results in significant voltage drops across the interconnects, and thus the current distribution across the crossbar is affected in a major way.

The Ta/HfO$_2$ crossbar has shape 128 × 64 and so this shape was chosen for all the simulations involving line resistance. Even relatively small ANNs of architecture 784(+1):25(+1):10 would need $2 \times (785 \times 25 + 26 \times 10) = 39,770$ memristors to be implemented. Even if not all the inputs were used at any given time, it would not be possible to fit all the memristors onto a single crossbar of shape 128 × 64. To overcome this, we decided to simulate multiple crossbars, each of which would implement a subset of the synaptic weights, but, for a given synaptic layer, would all compute in parallel. Because $\lceil 785/128 \rceil = 7$, seven crossbars were used to implement the first synaptic layer; the first crossbar utilized bottom 113 word lines, while the other six crossbars used bottom 112 word lines because $113 + 6 \times 112 = 785$. The second synaptic layer was implemented using eighth crossbar utilizing its bottom 26 word lines.

Figure 3a shows an example of how the first synaptic layer of 784(+1):25(+1):10 neural network could be implemented. Specifically, it shows how the first subset of weights would be implemented using one of the crossbars. Because we use proportional mapping scheme, positive and negative weights would be implemented in different bit lines. In Fig. 3a, memristors designated to implement positive weights are coloured in blue, memristors designated to implement negative weights are coloured in orange and unelectroformed memristors are coloured in black. Because simulations were constrained by experimental data, some of the devices were left unused and assumed to be unelectroformed. In practise, the crossbars could be manufactured to fit the geometry of the ANNs.

In each synaptic layer, the corresponding output currents from each of the crossbars would be added together. Additionally, output currents at the bit lines implementing negative weights

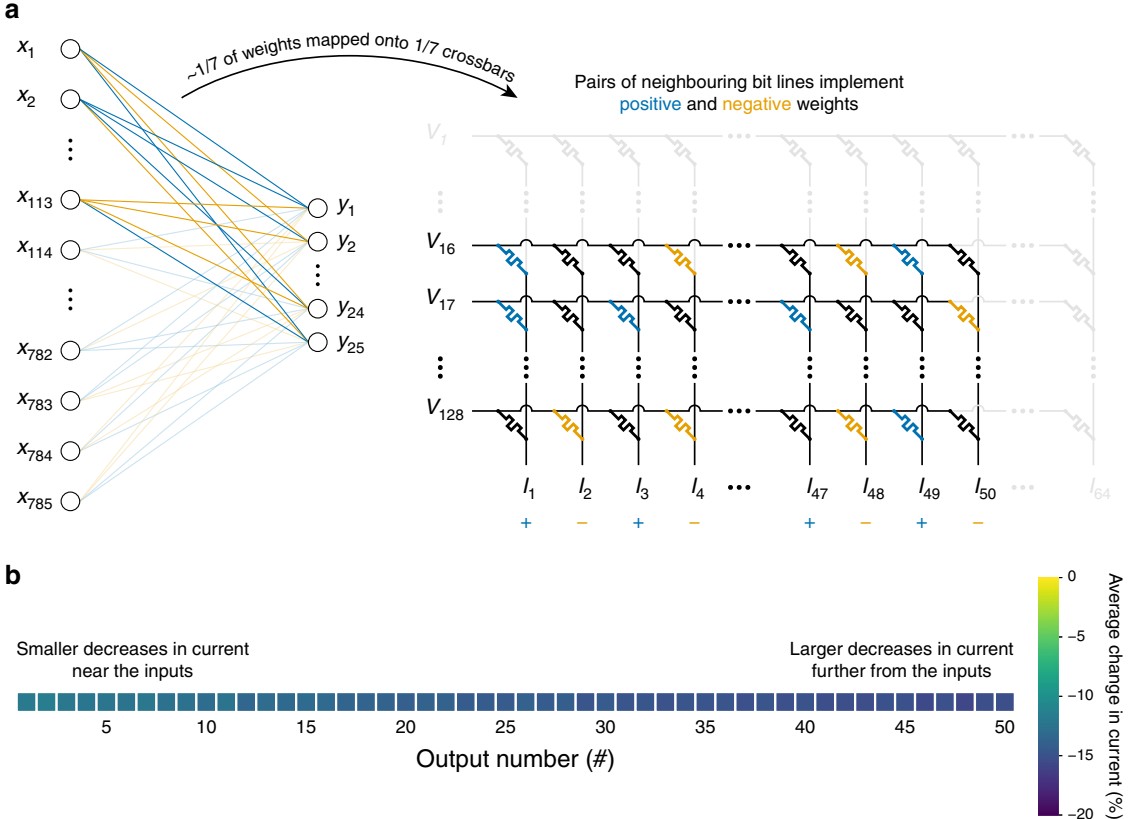

**Fig. 3 Theoretical implementation of a synaptic layer using crossbar arrays.** It was assumed that a synaptic layer of shape 785 × 25 would be implemented using seven crossbars of shape 128 × 64. **a** Mapping the first subset of weights onto one of the seven crossbars. Positive weights and negative weights are mapped onto memristors in different bit lines. **b** Heatmap of average changes in output currents due to line resistance (in all seven Ta/HfO₂ crossbars). For this particular simulation, it was assumed that Ta/HfO₂ devices can be programmed perfectly.

would be subtracted from the output currents at the neighbouring bit lines (to their left) implementing positive weights. For example, in the example configuration of Fig. 3a, output current at the 2nd bit line would be subtracted from the output current at the 1st bit line, etc.

Unfortunately, even when using multiple smaller crossbars, the interconnects can significantly disturb current distribution in the crossbar. Average output current decreases due to line resistance in all seven crossbars of Ta/HfO₂ devices (whose resistance ranges from ~1 kΩ to ~11 kΩ, and their interconnect resistance is 0.35 Ω and 0.32 Ω in the word and bit lines, respectively), are shown in the heatmap in Fig. 3b. We can see that the current decreases can range from ~12% at the outputs nearest to the applied voltages to ~16% at the outputs in the rightmost bit lines that are used. In Supplementary Note 1, we provide a possible strategy of mitigating line resistance effects in supervised learning. This scheme was not employed in the simulations described in the main text because we wanted to find out how well the CM method would deal with noticeable line resistance effects.

Inference accuracy: Figure 4 shows the accuracy of individual networks, as well as of their committees; memristive ANNs were simulated by taking into account three non-idealities of Ta/HfO₂ crossbar explored earlier—faulty devices, D2D variability and line resistance. As indicated by the yellow box plot in Fig. 4, individual networks implemented digitally achieve ~95.9% median accuracy. Networks disturbed to reflect the effect of non-idealities achieve ~91.0% median accuracy, as indicated by the vermilion box plot. Although that is a substantial drop in

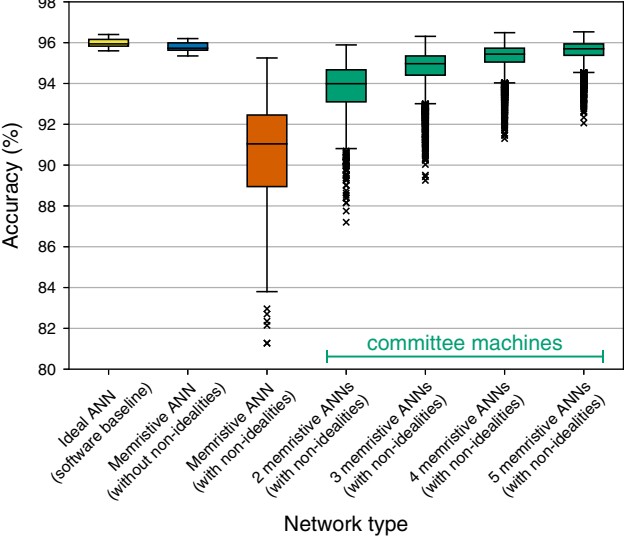

**Fig. 4 Accuracy of committee machines when using Ta/HfO₂ devices.** Accuracy achieved by individual networks and their committees when faulty devices, D2D variability data, and line resistance of Ta/HfO₂ crossbar are taken into account. The maximum whisker length is set to 1.5 × IQR.

accuracy, we see that as more networks are added to the committee, the more the accuracy increases. When five networks are used in a committee, median accuracy increases up to ~95.7%, as indicated by the rightmost green box plot.

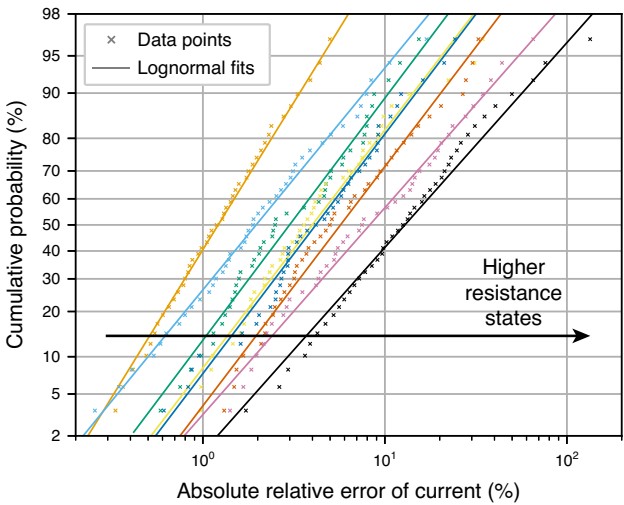

**Fig. 5 Random telegraph noise in a Ta₂O₅ device.** Cumulative probability plots of RTN-induced relative current deviations for all eight resistance states of a $Ta_2O_5$ RRAM device. Lognormal fits are shown for each resistance state.

**Ta₂O₅ RRAM**. In order to explore the effectiveness of minimising adverse effects of RTN, we use another memristor technology based on $Ta_2O_5$. To investigate RTN, measurements from a single device were considered. To simulate line resistance effects, interconnect resistance from $Ta/HfO_2$ was used and the same crossbar shape was assumed.

Random telegraph noise: Memristors often suffer from RTN resulting in a different accuracy at any given instant in time. $Ta_2O_5$ device was characterised by measuring the current of eight resistance states multiple times. Figure 5 shows the cumulative probability plots for those resistance states, together with lognormal fits modelling the nature of RTN. One of the things that the figure reveals is that higher resistance states suffer from higher degree of RTN. Fits for every resistance state, together with occurrence rates (see Supplementary Table 2), were used to disturb the weights of ANNs in order to reproduce the effect of RTN.

Inference accuracy: The results combining RTN and line resistance effects for $Ta_2O_5$ device are shown in Fig. 6. From the difference in median accuracy between yellow and blue box plots, we can notice that there is a significant drop in accuracy simply due to mapping of weights onto conductances. That is not surprising given that only eight states were available for mapping. One can also observe that further drop in median accuracy due to non-idealities is not as severe—it drops to ~94.1%. The RTN disturbance magnitude is limited to <100% in most cases, which possibly contributes to its smaller effect on accuracy. Additionally, $Ta_2O_5$ device has much higher resistance (ranging from 25 kΩ to 200 kΩ), thus line resistance is also less of a concern. When non-ideal networks are combined into committees of 5, the median accuracy jumps to ~96.5%—even higher than the software baseline of individual networks. This reveals additional trend seen in all the simulations performed—the higher the accuracy of the individual non-ideal memristive networks, the higher the accuracy of the committees that they are part of.

**aVMCO RRAM**. Further, we consider a third memristor technology—one based on aVCMO materials. We test the effects of RTN by considering measurements from a single device. Line resistance effects were simulated by using interconnect resistance and shape of $Ta/HfO_2$ crossbar array.

Random telegraph noise: Figure 7 shows the cumulative probability plots for eight resistance states of an aVMCO device

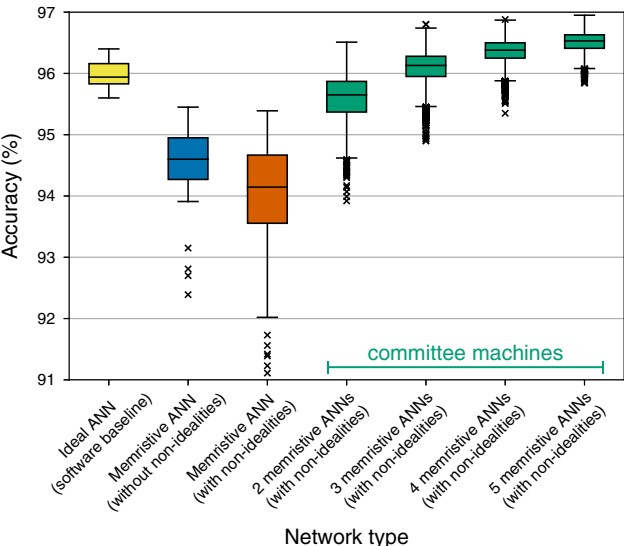

**Fig. 6 Accuracy of committee machines when using Ta₂O₅ devices.** Accuracy achieved by individual networks and their committees when RTN data of a $Ta_2O_5$ device are taken into account. Additionally, interconnect resistance of 0.35 Ω and 0.32 Ω in the word and bit lines, respectively, (from $Ta/HfO_2$ array) was used to include line resistance effects. The maximum whisker length is set to 1.5 × IQR.

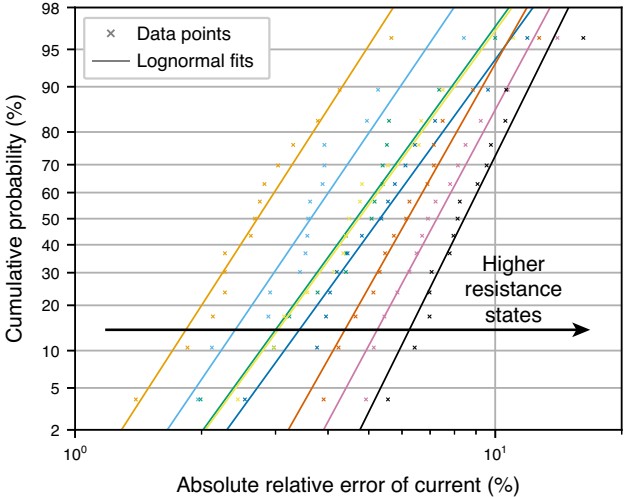

**Fig. 7 Random telegraph noise in an aVMCO device.** Cumulative probability plots of RTN-induced relative current deviations for all eight resistance states of aVMCO RRAM device. Lognormal fits are shown for each resistance state.

suffering from RTN. Like in $Ta_2O_5$, we observe that higher resistance states experience RTN of higher magnitude. However, compared to $Ta_2O_5$, the RTN magnitude is much more predictable. Fits for each of the eight resistance states, together with occurrence rates (see Supplementary Table 3), were used to simulate the effect of RTN in aVMCO-based neural networks.

Inference accuracy: The results combining RTN and line resistance are shown in Fig. 8. As with $Ta_2O_5$, we see a large drop due to mapping onto conductances—consequence of very few states available for mapping. More interestingly, the accuracy of individual memristor-based networks with and without non-idealities is almost identical. That is because the occurrence rate of RTN in aVMCO device is small and there is a much smaller

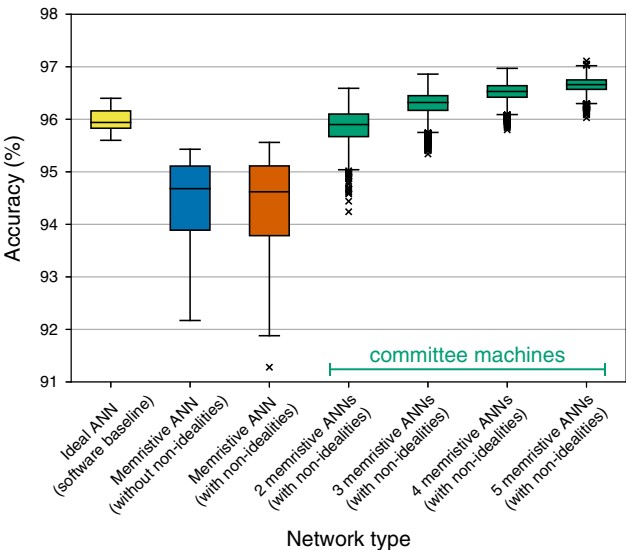

**Fig. 8 Accuracy of committee machines when using aVMCO devices.**
Accuracy achieved by individual networks and their committees when RTN data of an aVMCO device are taken into account. Additionally, interconnect resistance of 0.35 Ω and 0.32 Ω in the word and bit lines, respectively, (from Ta/HfO$_2$ array) was used to include line resistance effects. The maximum whisker length is set to 1.5 × IQR.

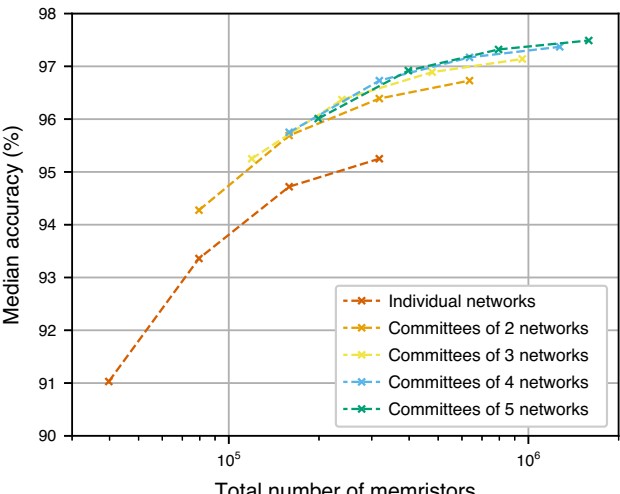

**Fig. 9 Effectiveness of committee machines when controlling for the total number of Ta/HfO$_2$ devices.** Median accuracy achieved by individual one-hidden-layer memristor-based networks and their committees. The networks contained 25, 50, 100, or 200 hidden neurons and were disturbed using faulty devices and D2D variability data from Ta/HfO$_2$ crossbar.

probability of RTN having large magnitude. Additionally, resistance of aVMCO device is even higher than that of Ta$_2$O$_5$ device—it ranges from 1 MΩ to 7.5 MΩ. Therefore, line resistance has even a smaller effect in a hypothetical array of aVMCO devices. Due to median accuracy of individual non-ideal memristor-based networks being higher (~94.6%), the median accuracy of committees is higher too—in committees of size 5 it increases to ~96.7%.

## Discussion

The results from the previous section suggest that the method of using committee machines to improve the accuracy of memristive neural networks is technology- and non-ideality-agnostic. CMs can mitigate the effects of faulty devices, D2D variability, RTN, and line resistance in combination with each other. Although CM method is slightly less effective with high-line resistance (see discussion in Supplementary Note 1), in all cases, we observe that the accuracy of individual non-ideal networks largely determines the accuracy of committees. That is consequential because it means that although committees always increase the accuracy, there is still an incentive to optimise the devices and systems that implement these networks—the higher the accuracy of individual networks, the higher the accuracy of the committees.

It is also important to consider whether using larger networks, instead of committees of smaller networks, would yield the same results if the same number of synapses (or memristors) was used in the large network as in the committee of smaller networks. In our previous work we found that the accuracy of networks before disturbance (which we call "starting accuracy") has a huge effect on the robustness to non-idealities—the larger the starting accuracy, the more robust the networks become[20]. One way to achieve higher starting accuracy is to have larger networks, e.g. if we have a network with one hidden layer, we might increase the number of neurons in that hidden layer, which would likely result in higher accuracy after training and thus higher robustness.

Figure 9 shows a comparison of CMs of memristor-based networks disturbed using faulty devices and D2D variability data from Ta/HfO$_2$ crossbar, when controlled for the total number of

memristors that is required to implement them (line resistance was not taken into account due to long time required to simulate it in large networks). We can observe that committees of two networks, each with 25 hidden neurons, (leftmost data point of the orange curve) achieve ~0.9% higher median accuracy than individual networks with 50 hidden neurons (second data point from the left in the vermilion curve), despite both requiring almost identical total number of memristors. Committees of two networks, each with 100 hidden neurons, (third data point from the left in the orange curve) achieve ~1.1% higher median accuracy than individual networks with 200 hidden neurons (rightmost data point in the vermilion curve), even though both require almost the same total number of memristors. Even larger improvement is gained when committees of four networks, each with 50 hidden neurons, (second data point from the left in the blue curve) are used instead—then the accuracy is improved by ~1.5%, with almost the exact total number of memristors used.

For different non-idealities and even different training schemes of the ANNs, the equivalents of Fig. 9 might be different, but there are a few common characteristics in all of them. In all cases, for a given total number of memristors used, there is an optimal number of networks that should be used in a committee. Additionally, we observe that the more severe a non-ideality is, the more apparent the effectiveness of committees becomes. Finally, sometimes the committees (for a fixed total number of memristors) might achieve lower accuracy than individual networks but only if the networks that they replace are very small and the non-ideality is not very detrimental. If the networks that are being replaced with committees of smaller networks, are sufficiently large, the committees will achieve higher accuracy. An example of that is shown in Supplementary Fig. 7 where aVMCO device is minimally affected by the non-idealities and so the advantage of committees becomes apparent only when replacing larger networks.

The reason why committees work in the context of non-ideal implementations and why they work best when they are used to replace large networks might, to some extent, lie in their training. When it comes to training fully connected networks, their accuracy tends to saturate as more parameters are added. Supplementary Fig. 4 shows that networks with 50 hidden neurons

can be trained to achieve significantly higher accuracy than networks with 25 hidden neurons. However, networks with 200 hidden neurons achieve only slightly higher accuracy than networks with 100 hidden neurons. This also means that networks with 200 hidden neurons will be only slightly more robust to non-idealities than networks with 100 hidden neurons. When such networks are affected by non-idealities, their accuracy drops to similar values but the smaller network can work in a committee with other networks, totalling almost the same number of memristors as the large network, but achieving higher accuracy overall. This is the most likely reason why the committees of smaller networks are effective at dealing with non-idealities, especially when replacing large networks.

In addition to the accuracy improvements, committees can provide flexibility in memristive implementations of neural networks. Digital implementations of ANNs have very predictable behaviour due to the precision of digital logic. Analogue implementations, on the other hand, can vary greatly even if they use the same weights before the mapping onto conductances—that is a result of the stochastic nature of memristors that implement these ANNs. The parallel and modular nature of committee machines makes memristive systems much more flexible. For example, if the verification accuracy of one of the ANNs in a memristor-based CM deteriorates below acceptable levels, its outputs could be disabled to ensure higher accuracy of the rest of the committee.

Importantly, this introduced parallelism comes at almost no extra cost. For a fixed total number of memristors, a committee of smaller networks, compared to a large individual network, would only require a few additional output and bias neurons, and an averaging functionality, which could potentially be implemented in hardware. For example, an ANN with 50 hidden neurons would require 846 neurons in total, while a committee of two ANNs, each with 25 hidden neurons (and thus requiring almost the same total number of memristors), would require 857 neurons in total.

In summary, our simulations employing experimental data from three different types of memristive devices show that committee machines employing ensemble averaging can be used to mitigate the effects of device- and system-level non-idealities in memristor-based neural networks. EA allows to achieve higher inference accuracy in physically implemented neural networks that suffer from faulty devices, device-to-device variability, random telegraph noise, and even line resistance. This method is a universal way to deal with the most common non-idealities and is straightforward to implement during the fabrication stage. Increased modularity of these memristive neural network systems will increase not only their inference accuracy, but also their robustness and flexibility, even without the need to sacrifice area. Although some level of non-idealities in memristors is unavoidable, CM method allows us to deal with these on the system level and is agnostic to a particular technology or, to some degree, type of the non-ideality.

## Methods

**Experiments.** Ta/HfO$_2$ RRAM 1T1R array consists of NMOS transistors fabricated in a commercial fab (feature size of 2 µm) and Pt/HfO$_2$/Ta devices. The bottom electrode was deposited by evaporation of 20-nm Pt layer on top of a 2-nm tantalum (Ta) adhesive layer; the electrode was patterned by photolitography and a lift-off process. A 5-nm HfO$_2$ switching layer was deposited by atomic layer deposition using water and tetrakis(dimethylamido)hafnium as precursors at 250 °C. Sputter-deposited Ta of 50 nm thickness followed by 10-nm Pd was used in a lift-off process to serve as the top electrode. The filamentary based Ta$_2$O$_5$ device consists of a TiN/4 nm stoichiometric Ta$_2$O$_5$/20-nm nonstoichiometric TaO$_x$/10-nm TaN/TiN stack with a cross-sectional area of 75 × 75 nm, while the non-filamentary-based aVMCO has a cross-sectional area of 135 × 135 nm and is composed of a TiN/8 nm amorphous-Si/8 nm anatase TiO$_2$/TiN stack. Ta$_2$O$_5$ and aVMCO devices were fabricated by imec. The detailed fabrication process parameters can be found in references[11,28,29] for Ta/HfO$_2$, Ta$_2$O$_5$ and aVMCO RRAMs, respectively.

The conductance of Ta/HfO$_2$ devices was modulated by applying SET pulses (500 µs @ 2.5 V and gate voltage increasing from 0.6 to 1.6 V). After each of the 11 cycles, RESET pulses were applied (5 µs @ 0.9 V increasing to 2.2 V and gate voltage of 5 V). The voltage was being increased linearly throughout the 100 pulses. All electrical tests for Ta$_2$O$_5$ and aVMCO devices were done with a Keysight B1500A. The RTN data are extracted by switching the device into eight uniformly distributed resistance levels between 25 kΩ and 200 kΩ, and eight nearly uniformly distributed resistance levels between 1 MΩ and 7.5 MΩ with incremental RESET DC sweeps[30] for Ta$_2$O$_5$ and aVMCO, respectively. RTN measurement is then carried out at each resistance level at a 0.1 V and 3 V read-out for Ta$_2$O$_5$ and aVMCO respectively, with a sampling time of 2 ms/point and 10,000 sampling point per resistance level for an RTN measurement period of 20 s.

**Simulations.** In this work, feed-forward ANNs with fully connected layers and continuous weights were trained to recognise handwritten digits using the MNIST data base. All 60,000 MNIST training images were used during the training stage; training set consisted of 50,000 images and verification set consisted of 10,000 images. All 10,000 test images were used to evaluate the inference accuracy of ANNs. Networks used 784 input neurons representing pixel intensities of MNIST images of 28 × 28 pixel size, as well as one bias neuron. Ten output neurons were used; they represented the ANNs' predictions of 10 handwritten digits. Hidden layers used sigmoid activation function, while the output layer used softmax activation function. Weights were optimised by minimising cross-entropy error function using stochastic gradient descent. Learning rate of 0.01 and patience of 25 epochs were used. Twenty-five networks were trained for each architecture explored by initialising them differently. When numerically optimising ANNs' weightings, optimisation was performed by employing verification set, while the performance was evaluated using the test set. The code was implemented in Python.

Weights were mapped onto pairs of memristors' conductances using proportional mapping scheme—synaptic weights were made proportional to one of the conductances in the pair, while the other was left unelectroformed. The zero weight was interpreted as given—in practise, it would be implemented by not electroforming the device, thus resulting in its negligible conductance. Although aVMCO devices do not have electroforming stage, for consistency we assumed that additional insulating circuit elements could be used to implement the zero weight. Negative weights would be implemented by placing certain memristors in dedicated bit lines of the crossbars whose outputs would be subtracted from the outputs at the corresponding bit lines implementing positive weights. Maximum weights after mapping were optimised separately for each set of network architecture and conductance levels; in each case this was done by excluding a certain proportion, $p_L$, of weights with largest absolute values. What $p_L$ values were used for each simulation is summarised in Supplementary Table 1. More details on the mapping procedure can be found in our past work[20].

All non-idealities, except for line resistance, were simulated by disturbing the individual conductances of memristor-based ANNs. To investigate line resistance, nodal analysis was employed. By setting up simultaneous linear equations using Ohm's law and Kirchhoff's current law, those were solved in sparse matrix representation using Python's library scipy.

After simulating memristor non-idealities, committees of different ANNs were composed. Committees used EA, i.e. the outputs of individual networks in a committee were averaged to produce a single output vector. In EA, the output vectors of individual networks can simply be added together (if the weightings of different networks are the same, as we assume in the main text); the label corresponding to the entry with the highest value would be the prediction of the committee. This addition can be performed either in software, or, if the activation function of the last neuronal layer can be implemented physically, it can be performed by adding corresponding currents produced by the circuitry of this activation function.

In the simulations, neural networks that go into a committee were chosen randomly. This was done to reflect the most convenient strategy when manufacturing such systems—because one does not need to selectively choose the networks, manufactured crossbars can be easily programmed without the need to replace them if they perform poorly when working individually (unless their effect is so detrimental that they have to be ignored, which can be made possible with this technique). Besides, devices might change over time, thus these simulations, which show what happens when one does not selectively choose the networks, are valuable to investigate conditions where it is not possible to replace the networks.

In the simulations, 25 base networks were used (each having different set of weights) for each of the architectures. Then all of their weights were mapped onto pairs of conductances using HRS/LRS values extracted from experiments. Finally, to reflect the effect of each of the non-idealities, all networks were disturbed multiple times. In each disturbance iteration, multiple combinations of networks were chosen and their performance as a committee of certain size was evaluated. In total, for most simulations, 10,000 data points were recorded for a committee of every size—these data captured the variations of base networks, their combinations and different disturbance iterations. Only simulations involving line resistance or numerical optimisation of weights had fewer data points for some committee sizes (due to long simulation times).

## Data availability

The experimental data that support the findings of this study are available from the corresponding author upon reasonable request. Data generated during the simulations are provided as a Source Data file. Source data are provided with this paper.

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

## Acknowledgements

A.M. acknowledges funding from the Royal Academy of Engineering under the Research Fellowship scheme, A.J.K. acknowledges funding from the Engineering and Physical Sciences Research Council (EP/P013503/1) and the Leverhulme Trust (RPG-2016-135), W.D.Z. acknowledges funding from the Engineering and Physical Sciences Research Council (EP/S000259/1), D.J. acknowledges studentship funding from the Engineering and Physical Sciences Research Council (ref. 2094654).

## Author contributions

A.M. and D.J. conceived the idea and designed the study. P.F. and Z.C. performed the experimental measurements. D.J. performed the simulations and analysed the experimental and simulation results. C.L. and Q.X. provided the experimental data of the programming of a Ta/HfO$_2$ 1T1R RRAM array. W.H.N. and M.B. contributed to the understanding of the electrical behaviour of memristor devices. A.M., W.D.Z., and A.J.K. supervised the research. D.J. wrote the initial manuscript. All authors contributed to the discussions of the results and improved the text.

## Competing interests

The authors declare no competing interests.
