## [Peer Review File · Nature Communications]

Reviewers' comments:

Reviewer #1 (Remarks to the Author):

It's an interesting study to bring the idea of ensemble averaging to memristive neural networks, trying to overcome some non-idealities of the devices and the structure. The idea is to form a committee of different disturbed networks due to the non-ideality from identical or different continuous networks, and use the averaged output of all the networks as the final output. The simulation model for several non-ideal factors are extracted from experimental data from 3 different types of devices. The math and statistics shown are clear and the improvement of the proposed method is effective. Overall, I consider this work very useful for the community trying to use memristor networks to obtain decent computation accuracy. I recommend it for publication after the following minor issues are addressed:

1. As the authors assumed in the manuscript, one may argue that this technique requires multiple networks working together and thus multiple times of area and power. Although the authors answered this question in Figure 5B, which only considered the disturbance using C2C data from TaOx devices. Is it possible to consider other disturbance to strengthen the claim that ensemble averaging can always help due to the saturation of a single network? Besides, there are previous simulation works considering multiple types of device non-idealities simultaneously.
2. Following up with the need of using multiple networks for this approach, the authors may want to comment on combining it with weight sharing approach (Nature Machine Intelligence 1, 434 (2019)) to minimize the hardware resources while maintaining a decent accuracy?
2. In line 123, it is mentioned that sometimes the physical networks can achieve even higher accuracies than that of individual continuous networks. Is there any short explanations if all the physical versions are just inferior versions of their continuous correspondence? Or am I misunderstanding?
3. Are all the experimental data shown in Figure 2 obtained from only one device as implied in methods?
4. Random reordering is a good idea against the line resistance effect. However, people may be curious about the raw performance without it. It could be put in the supplementary information.
5. In simulation methods, implementing zero weights by not electroforming the device is not always practical, especially when the reconfigurability is considered.

Reviewer #2 (Remarks to the Author):

The authors present results on inference accuracy for different neural networks model implementing memristive devices, in particular Resistive Random Access Memories (RRAM). Their work is focused on increasing the accuracy of classification utilizing ensemble on neural networks in parallel, instead of a single large network. The work consist in simulation based on measured device non-idealities, such as Random Telegraph Noise (RTN), cycle to cycle variability, and device to device variability. The results shows an accuracy improvement utilizing an ensemble of networks, dubbed Committee Machines.

The reviewer thinks that the concept is interesting, however array level experiments are needed to have a fair comparison with other works. The results cited by the authors are usually based on array level experiment, wether this paper is based on single device experiments followed by array simulations. A part from this, here's a few suggestion to improve the quality of the paper in the reviewer opinion:

1. It is clear that the authors want to show that their concept is technology agnostic. However it is confusing to see and compare results from different technologies all in once. The suggestion is to try to re-organized the paper such that a single technology is taken into account (example SiOx), characterized, and the results are simulated just for that technology, Then, the results are extend to others showing just a brief comparison in a final figure. Otherwise it ends up with not very well organized paper

2. Many of the results were already presented in literature (example, the effect of non-linearity). In particular you can find some interesting results in G.Burr et. al., TED 62(11)
 3. The conduction non linearity actually is not crucial for inference. In fact, program and verify techniques are always performed.
 4. How do you compare these results with in-situ training (which is already 'aware' of defects and non linearity)?
 5. Since the ensemble network has more neurons compared with the singles one, how does the power consumption scale?what about latency? How much parallelism can be achieved?
 6. The presented network is a simple architecture, how does the committed machines concept apply to deep network, for example to Convolutional Neural Networks?
- At this stage, the reviewer doesn't think that the paper is suitable for nature communication, at least a small array demonstration is needed.

Reviewer #3 (Remarks to the Author):

The paper is well written and the results are clear. The paper, however, is lacking in novelty. Using averaging techniques to deal with hardware nonlinearities is hardly new (see, for instance, the old "common centroid" method of laying out critical components to average across various non-idealities. Here, often a large sized transistor is divided into smaller transistors so that the non-ideal influences are balanced across the components.) In what you have presented here "committees" are similar to averaging and if I look at Fig. 5 B, we can see that adding more "committees" doesn't improve the results much, beyond 95% accuracy, which, on a simple dataset like MNIST is very bad performance. In the paper you eluded to other ways in which committee machines could be emulated beyond ensemble averaging, and I think to be novel, this paper needs to explore those ideas further.

Major Changes to the Manuscript

We thank the reviewers for insightful comments and suggestions. Following that, we have made a large number of additions to the manuscript, including new experimental results, new simulations, as well as further explanations or clarifications. We believe that the additions addressed all reviewers' comments and significantly improved the manuscript by making the arguments stronger and the evidence more complete. There are a large number of changes to the manuscript that can be found in the second parts of the attached documents, containing tracked changes. Major changes (i.e. those that include addition of new experimental and/or simulation data) are listed below, while responses to individual comments can be found in the rest of the document. Unless stated otherwise, any mentions of figures, tables or line numbers refer to the newest version of the manuscript.

1. Conducted and included array-level experiments of Ta/HfO₂ devices and performed additional simulations using the new data (see Figures 2, 4D and S15). We performed an extensive experimental analysis of 128 × 64 1T1R Ta/HfO₂ RRAM-based crossbar array by analysing 8192 RRAM devices over multiple programming cycles.
2. Performed additional simulations exploring numerical optimisation of ANNs' weightings in committee machines (CMs). Although even simple averaging allowed significant increases in accuracy, we decided to explore whether it would be possible to improve this method further. Optimising weightings for each separate ANN yielded marginally better results which were presented in the supplementary information (see Figure S11).
3. Additional results exploring the effectiveness of CMs when controlling for size were added to supplementary information (see Figures S12-S17). They show that although the size of the effect differs for different non-idealities, CMs improve the accuracy in all cases. Even in the worst performing case (large proportion of devices stuck at LRS), where CMs seem to not perform as well when the networks they replace are small, the increase in accuracy when replacing larger networks is very significant.
4. Additional results exploring the effect of line resistance on accuracy (when no reordering of inputs or outputs is used) have been added to supplementary information (see Figure S1). Although that is not the central point of the paper, the results show that when no reordering is used, the line resistance effects are extremely detrimental.

Responses to Individual Comments

Reviewer 1

It's an interesting study to bring the idea of ensemble averaging to memristive neural networks, trying to overcome some non-idealities of the devices and the structure. The idea is to form a committee of different disturbed networks due to the non-ideality from identical or different continuous networks, and use the averaged output of all the networks as the final output. The simulation model for several non-ideal factors are extracted from experimental data from 3 different types of devices. The math and statistics shown are clear and the improvement of the proposed method is effective. Overall, I consider this work very useful for the community trying to use memristor networks to obtain decent computation accuracy. I recommend it for publication after the following minor issues are addressed:

We thank the reviewer for constructive comments, and we are encouraged by the opinion that this would be a valuable result for the community. Please see the responses to specific questions below.

1. As the authors assumed in the manuscript, one may argue that this technique requires multiple networks working together and thus multiple times of area and power. Although the authors answered this question in Figure 5B, which only considered the disturbance using C2C data from TaOx devices. Is it possible to consider other disturbance to strengthen the claim that ensemble averaging can always help due to the saturation of a single network? Besides, there are previous simulation works considering multiple types of device non-idealities simultaneously.

We observe the improvement in accuracy for all non-idealities. The increase in accuracy can differ slightly for different non-idealities, but generally the committees perform better. To demonstrate this, we have now added equivalents of Figure 5B (old version) for all non-idealities, except line resistance, in the supplementary information (see Figures S12-S17). We have not included line resistance because accurate simulations of large crossbars arrays would require extremely long time to complete.

2. Following up with the need of using multiple networks for this approach, the authors may want to comment on combining it with weight sharing approach (Nature Machine Intelligence 1, 434 (2019)) to minimize the hardware resources while maintaining a decent accuracy?

The simplicity of ensemble averaging means that it can be (and was) applied to more complex neural networks, such as convolutional neural networks (CNNs). The paper, that reviewer references, shows how CNNs can be implemented physically using crossbar arrays. The reviewer thus correctly notes that our approach could be applied to techniques employing weight sharing (i.e. CNNs). We have now mentioned this in the main text and referenced the suggested paper (lines 141-147):

”Although in this work we focus on fully connected ANNs, CMs could be applied to other variants of neural networks as well. Due to the simplicity of EA, it could, for example, be employed in convolutional neural networks (CNNs) [23], which are often used for image classification. This might be of interest as CNNs have been successfully implemented using crossbar arrays recently [24]. However, crossbar implementations are naturally more suited to fully connected networks, therefore we limit ourselves to this architecture but are open to exploring the effectiveness of EA with CNNs in the future.”

We also note that CNN implementations include fully connected layers in addition to convolutional layers. For example, AlexNet has five convolutional layers with 2.3 million weights, and three fully connected layers with 58.6 million weights. Although in more recent implementations of CNNs the number of convolutional layers has increased, the total number of weights in fully connected layers still represents a significant portion of the total number of weights and thus consumes a significant amount of energy.

3. In line 123, it is mentioned that sometimes the physical networks can achieve even higher accuracies than that of individual continuous networks. Is there any short explanations if all the physical versions are just inferior versions of their continuous correspondence? Or am I misunderstanding?

That is exactly correct. One can think of disturbed networks (which are the models of the physical implementations) as being inferior versions of the continuous networks. Because the nature of non-idealities is more or less random, it is most likely that disturbed networks will perform worse than their continuous counterparts. However, because we use different continuous networks for different physical implementations, when combined, they can outperform individual continuous networks.

4. Are all the experimental data shown in Figure 2 obtained from only one device as implied in methods?

Initially, we characterised individual devices to explore the nature of different non-idealities. Original figures represented typical results obtained from single devices. In the revision of the manuscript, we have significantly expanded our study to include a 128×64 Ta/HfO₂-based crossbar array (see Figure 2A) and also to include device-to-device variability. We analysed 8192 devices, each being subjected to multiple pulsing cycles (11 cycles per device). The experimental results for these devices are shown in Figure 2B, and we moved some of the original data to the supplementary information. In total, we have analysed four different technologies—SiO_x-based RRAM, Ta₂O₅-based RRAM, aVCMO-based RRAM and Ta/HfO₂-based RRAM. The full Figure 2 is shown below:

5. *Random reordering is a good idea against the line resistance effect. However, people may be curious about the raw performance without it. It could be put in the supplementary information.*

That is a valid point and we agree with the reviewer. We have now added these results in Figure S1 of supplementary information. The effectiveness of committees when dealing with interconnect resistance of $2.5\ \Omega$ is shown below:

6. *In simulation methods, implementing zero weights by not electroforming the device is not always practical, especially when the reconfigurability is considered.*

That is correct. One option is to use the memristor differential pairs. However, when it comes to inference, there is no unique way of assigning conductances of two memristors in a differential pair to represent a synaptic weight, i.e. there is an infinite number of possibilities and the optimal choice might depend on the nature of variability in a particular type of devices. Thus, for simplicity, we use

proportional mapping scheme, which implements zero weights by not electroforming the devices. We believe that regardless of the way the zero weights are implemented, the qualitative results presented in the paper (inference accuracy improvements) would not change.

Reviewer 2

The authors present results on inference accuracy for different neural networks model implementing memristive devices, in particular Resistive Random Access Memories (RRAM). Their work is focused on increasing the accuracy of classification utilizing ensemble on neural networks in parallel, instead of a single large network. The work consist in simulation based on measured device non-idealities, such as Random Telegraph Noise (RTN), cycle to cycle variability, and device to device variability. The results shows an accuracy improvement utilizing an ensemble of networks, dubbed Committee Machines. The reviewer thinks that the concept is interesting, however array level experiments are needed to have a fair comparison with other works. The results cited by the authors are usually based on array level experiment, wether this paper is based on single device experiments followed by array simulations. A part from this, here's a few suggestion to improve the quality of the paper in the reviewer opinion:

We thank and fully agree with the reviewer that array-level experiments can make a more compelling case for using committee machines to mitigate the effects of non-idealities in physically implemented neural networks. Following this suggestion, we have now included in our analysis experimental results from 128×64 Ta/HfO₂ crossbar array, which is shown in Figure 2A. To our knowledge, this is one of the largest experimentally demonstrated RRAM-based arrays. The data used explore the effect of non-linear programming, but the results also capture the effect of faulty devices (stuck in one state) and, naturally, device-to-device variability. We analysed 8192 devices by performing multiple pulsing cycles (11 cycles per device). The experimental results from these devices are presented in Figure 2B. The full figure is shown below:

Simulation results involving these results show—as with other types of non-idealities—that committee machines can significantly increase the accuracy of non-ideal networks (see Figure 4D), even without increasing the number of devices, as shown in Figure S15:

1. It is clear that the authors want to show that their concept is technology agnostic. However it is confusing to see and compare results from different technologies all in once. The suggestion is to try to re-organized the paper such that a single technology is taken into account (example SiO_x), characterized, and the results are simulated just for that technology, Then, the results are extend to others showing just a brief comparison in a final figure. Otherwise it ends up with not very well organized paper

Again, we fully agree and appreciate this suggestion and have now separated experimental results from different technologies into different figures. Ta/HfO₂ results are now presented in Figure 2B, Ta₂O₅ results are presented in Figure 3 and SiO_x results have been moved to supplementary information to Figure S2. In total, we have analysed four different technologies—SiO_x-based RRAM, Ta₂O₅-based RRAM, aVCMO-based RRAM and Ta/HfO₂-based RRAM.

2. Many of the results were already presented in literature (example, the effect of non-linearity). In particular you can find some interesting results in G.Burr et. al., TED 62(11)

We thank the reviewer for the suggestion. We have now included a brief description of the importance of linear conductance response (especially in in-situ training) and referenced the mentioned paper (lines 159-163):

“More elaborate programming schemes use pulses with changing amplitude or width, but not even that can guarantee perfectly linear modulation of conductance. Although this is more relevant for in-situ training where it is necessary to ensure accurate adjustment of ANN’s weights [25], linear conductance modulation is preferable even for inference.”

3. *The conduction non linearity actually is not crucial for inference. In fact, program and verify techniques are always performed.*

Among the two types of non-linearity (programming and I/V non-linearity), the programming non-linearity is definitely a more significant issue (at least for in-situ training). Although the verify techniques are almost always used, the complexity of these would have significant implications on energy efficiency. The more linear the response is, the less complicated the programming schemes have to be; this results in less additional circuitry needed and energy/time savings.

We note this is one of many types of non-idealities we consider. Even if we can completely ignore programming non-linearity, the newly introduced results (see Figure 2B) also show the effect of stuck devices and device-to-device variability. Thus, we believe, they are worthy of exploring, in addition to previously analysed non-idealities.

4. *How do you compare these results with in-situ training (which is already 'aware' of defects and non linearity)?*

It is difficult to compare our results with in-situ training demonstrations as there is no straightforward way to use ensemble averaging for training. In-situ training might be able to deal with some non-idealities by minimising the cost function while taking the effect of the non-idealities into account in real time. However, in our case, we believe there would be significant issues with adjustment of the weights (device conductances), and that it might take a long time for training to converge to the acceptable levels of accuracy. Typically a training phase requires a much higher resolution of weights (FP32/16 is used in digital implementations). There have been a number of excellent works examining hybrid approach—combining CMOS and memristive technology for training to mitigate this issue, for example [7] and [16]. In-situ training is outside of the scope of this study.

We have now added additional reference to highlight this in the text (line 57).

5. *Since the ensemble network has more neurons compared with the singles one, how does the power consumption scale? what about latency? How much parallelism can be achieved?*

The answer to this question might, to some extent, depend on the crossbar size and how many synaptic layers could be arranged in it. However, we do not think that extra neurons would produce problems.

If we replace a large ANN consisting of one hidden layer with a set of smaller ANNs (each with one hidden layer), the number of neurons will essentially only increase in the input and output layers. Although each ANN requires its own input neurons, in a committee machine each network would receive the same inputs so it is often possible to arrange first synaptic layers of each ANN next to each other in a crossbar array. When it comes to output neurons, there is usually much fewer of them—in the case of MNIST data set, there are two orders of magnitude fewer output neurons than there are input neurons. Thus, we believe that this would not have a significant effect on the efficiency of the system.

When it comes to parallelism, there essentially is no theoretical limit—only the available crossbar sizes determine the number of ANNs one can implement. Because all the ANNs in a committee machine compute in parallel, the latency is not affected. The only additional operation that needs to be performed is averaging, which, if the activation function of the last neuronal layer can be realised in hardware, can too be performed using analogue circuitry.

6. The presented network is a simple architecture, how does the committed machines concept apply to deep network, for example to Convolutional Neural Networks?

Committee machines and, specifically, ensemble averaging have been applied to convolutional neural networks (CNNs) before[23]. It is also possible to implement CNNs using crossbar arrays. Thus, one could also employ the concept of ensemble averaging to physically implemented CNNs. Although we limit ourselves to fully connected networks in this analysis, we have made an addition to the main text noting this possibility (lines 141-147):

“Although in this work we focus on fully connected ANNs, CMs could be applied to other variants of neural networks as well. Due to the simplicity of EA, it could, for example, be employed in convolutional neural networks (CNNs) [23], which are often used for image classification. This might be of interest as CNNs have been successfully implemented using crossbar arrays recently [24]. However, crossbar implementations are naturally more suited to fully connected networks, therefore we limit ourselves to this architecture but are open to exploring the effectiveness of EA with CNNs in the future.”

We also note that CNN implementations include fully connected layers in addition to convolutional layers. For example, AlexNet, has five convolutional layers with 2.3 million weights, and three fully connected layers with 58.6 million weights. Although in more recent implementations of CNNs the number of convolutional layers has increased, the total number of weights in fully connected layers still represents a significant portion of the total number of weights and thus consumes a significant amount of energy.

At this stage, the reviewer doesn't think that the paper is suitable for nature communication, at least a small array demonstration is needed.

Once again, we thank the reviewer for the careful reading and very constructive comments on how to improve our manuscript. By adding the analysis of RRAM-based crossbar array containing 8192 devices, as well as answering your validly raised concerns, we hope that the reviewer finds the current version suitable for the publication in Nature Communications.

Reviewer 3

The paper is well written and the results are clear. The paper, however, is lacking in novelty. Using averaging techniques to deal with hardware nonlinearities is hardly new (see, for instance, the old "common centroid" method of laying out critical components to average across various non-idealities. Here, often a large sized transistor is divided into smaller transistors so that the non-ideal influences are balanced across the components.) In what you have presented here "committees" are similar to averaging and if I look at Fig. 5 B, we can see that adding more "committees" doesn't improve the results much, beyond 95% accuracy, which, on a simple dataset like MNIST is very bad performance. In the paper you eluded to other ways in which committee machines could be emulated beyond ensemble averaging, and I think to be novel, this paper needs to explore those ideas further.

Firstly, we would like to thank the reviewer for the useful comments and interesting points raised.

We can see the similarities between our CM approach and common centroid method described by the reviewer. However, we also see some significant differences between the two approaches. To the best of our understanding, the common centroid method for laying out critical components is used to mitigate the non-idealities between typically a pair of critical components (e.g. transistors) where the negative effects cancel out. In contrast to this method, in the case of CM, we utilise the differences between different members of the committee. For example, if all ANNs in the ensemble are identical, the approach would not yield any improvement in the performance. Additionally, the CM approach is a system-level solution that is, to a large extent, agnostic to technology or type of non-ideality and thus does not require any careful alignment of components. Furthermore, one of the most challenging aspects of memristive technologies is the inherently stochastic nature of the switching process. It is very hard to determine the trends of variability (such as trends in cycle-to-cycle variability or RTN conductance deviations). The common centroid method for laying out critical components might be much more challenging when memristors are used instead of CMOS components. The variability in RRAM devices is not only the result of the fabrication process but also of the intrinsic stochasticity of ionic movement that controls the resistance switching. To our knowledge, there are no studies that directly consider using common centroid method to mitigate the non-idealities of memristive devices. Although methods like common centroid are an interesting area of exploration, we do not believe that the memristor physics are understood to the point where these methods could be as successfully utilised as in the case of mature CMOS technologies. We thus aim for system-level approaches that do not require complete understanding of device physics and the nature of their stochasticity.

However, we believe that in principle, it would be beneficial to combine the two approaches, common centroid and CM. It is similar to the way the materials' optimisation of memristive technologies could go together with the CM approach. We see the CM as a system-level solution that could be combined with any other technique that improves the performance of individual devices or critical components.

In this paper we focus specifically on analogue implementations of ANNs, where averaging has not been explored properly (if hardly at all). The reason we believe in the importance of this work is because of how much time in the neuromorphic community is being spent on device optimisation (see Table I), while very few alternatives are explored. Although committee machines are not a panacea for all the challenges that RRAM-based neural networks face, they provide a simple, easily implementable way of improving accuracy with essentially no trade-offs and which can be combined with other device-level improvements.

We appreciate your suggestion to further explore the different CM approaches. Following this, we have now considered more elaborate committee machines. We believe, however, that when it comes to inference (as opposed to in-situ training), ensemble averaging works best. Nevertheless, even within the realm of this type of committee machines, there is the possibility to explore different weightings for each network. One of the oldest of such techniques is generalized ensemble method (GEM). Although it provides an optimal way of determining the weightings for each ANN, it did not suit our scenario due to a different cost function used (lines 102-104):

“But because [20] only considered networks with mean square error loss function (while our networks used cross-entropy loss function), this work does not explore GEM.”

We have thus instead explored numerical optimisation of ANNs’ weightings. The results are only marginally better than those employing simple averaging, so we presented them in the supplementary information only. However, it shows that there are ways to slightly improve even this simple method. The results of committee machines with numerically optimised weightings dealing with the effects of C2C variability are presented in Figure S11:

With regards to relatively low accuracy achieved in some of the simulations, our goal was not to achieve state-of-the-art accuracy, but rather to show how to improve it in any scenario. If we compare our results with state-of-the-art convolutional neural networks, then the accuracy of physically implemented ANNs indeed does not look impressive. However, we believe we should compare our results to those within neuromorphic community.

The main goal of implementing ANNs physically is to drastically reduce the power consumption. This, with the current state of neuromorphic technology, unfortunately comes at a cost of reduced accuracy. This paper aims to show how the accuracy can be improved regardless of non-ideality or the network architecture used. We did not selectively pick results, so C2C disturbance (in Figure 5B (old version)), in the case of Ta₂O₅ devices, was rather severe. This resulted in absolute accuracy of the committees not being very high. However, the improvement of ~5.4% by going from individual network with 240 hidden neurons to 4 networks with 60 hidden neurons (i.e. same total number of synapses) is very significant.

The absolute accuracy of a committee can be improved by having higher accuracy of the networks that it consists of. That is the reason why system-level approaches, like ensemble averaging, will be only part of the solution and having well-optimised devices will still be important. We also analysed other non-idealities whose effect is not as detrimental and have now included additional results in the supplementary information. For example, Figure S16 shows the effectiveness of committees when dealing with RTN, whose negative effect is not as large as that of C2C:

We can see that in that case, not only is the relative accuracy increase significant, but also that the absolute accuracy achieved is much higher (~98% in larger committees) and thus could be sufficient for certain applications.

Finally, we thank the reviewer for the comments. We have now carefully considered reviewer's comments and conducted additional analysis and simulations. We hope that the revision of the manuscript is now acceptable for the publication in Nature Communications.

Reviewers' comments:

Reviewer #1 (Remarks to the Author):

The authors have addressed my previous questions and significantly improved the manuscript, which is essentially acceptable now. A remaining minor concern I have is that Figure 2A in the revised manuscript has been published before in previous publications of some of the co-authors. The authors may want to cite these previous publications in the figure caption to avoid any possible confusion.

Reviewer #2 (Remarks to the Author):

The authors revised the manuscript adding measurements also on array level, which is highly appreciated by this reviewer. Overall the topic is interesting but the paper organization in my opinion lacks of coherence. The major issues are summarized in the following points that were also partially raised during the previous review stage:

1- I understand that multiple networks work in parallel, but making things work in parallel it's not easy. It is ok to state at line 68 (introduction) that is not a problem, but the authors should present an estimate of the resource needed later on the main text. Even if the number of weights does not change one has to acquire multiple outputs (it's not possible to have so many ADCs for example), compute the neuron activation and process it to the next neural network layer. At the end the ensemble has to be computed. How this would impact the latency and energy performance?

2- Non-linear programming. The authors haven't still convinced me that non-linear programming is an issue for inference. Programming is a one-time operation for an inference machine, so in principle one could spend a lot of time programming the device with precise program and verify scheme. How the non-linearity simulation were performed? Did any program and verify algorithm were included? In the reviewer opinion, non-linear programming has nothing to do with a precise inference and it is out of the scope of this paper. The authors might present another work where training is taken into account and in that case non-linearity has a huge impact on performances.

3- Same thing apply for cycle to cycle (C2C) variability. Why should it be a problem for inference? I have to program the device only one time. In the reviewer opinion also the C2C consideration should be removed from this work and kept for a second work focused on training

4- In general the organization of the paper is confusing: Non linear programming is simulated with data from Ta/HfO2 array and SiOx device, while RTN is simulated with Ta2O5 and aVCMO data and C2C variability with only Ta2O5 data. The reviewer thinks that a correct benchmark should be obtained with all the data from all the technology.

To have an high impact and quality paper I suggest to re-organized the paper as follow

- 1- present the commitment machine
- 2- measure at array level in the Ta/HfO2 array the commitment machine accuracy
- 3- measure at array level in the Ta/HfO2 array a neural network with the same number of weights as (2) and compare the results
- 4- measure and characterize RTN on the Ta/HfO2 array and estimate the parasitic interconnect resistance, measure and characterize RTN in SiOx, Ta2O5, aVCMO devices and simulate the effect of RTN and interconnects for commitment machines and simple neural networks, comparing different technology showing that the technique is technology agnostic.
- 7- remove completely the effect of non-linear programming and C2C variability

Right now it seems that the authors have used data from different technologies and groups

without a coherent characterization procedure that should be the same and extract the same type of data for all the technology to have a fair comparison.

I understand that this is a lot of work and it looks like a complete new paper, but the authors have in their hands a very powerful tool that if well organized and presented can attract strong interest from the community and high impact. In fact RTN (and other noise issues) and parasitic interconnects resistance drops are huge problems that need to be solved, and I think that commitment machine solution is very promising but should correctly be characterized and presented.

For these reasons, the reviewer thinks that while the work is very interesting it should be reorganized before publication on Nature Communications

Reviewer #3 (Remarks to the Author):

The changes that the authors have made have improved the quality of the paper. The results of the network are still poor, however, and I would urge the authors to try another, more relevant, dataset besides MNIST. By trying another dataset you might be able to demonstrate further advantages of your technique.

The paper is very well written and the images are clear.

Major Changes to the Manuscript

We thank the reviewers for the suggestions. To improve the quality of the manuscript, we have redone all simulations and took a more structured approach to the analysis of non-idealities and technologies. The most important changes are listed below, while the rest of the changes in the main text can be found in the second part of the attached document, as well as in the individual responses to the reviewers in the following pages. Supplementary information document was redone completely so the attached document does not contain its tracked changes.

1. For all technologies, combined non-idealities whose data were available. For Ta/HfO₂ devices, we combined faulty devices and D2D variability data, as well as line resistance. For the Ta₂O₅ and aVMCO devices, we combined their RTN data and used interconnect resistance from Ta/HfO₂ to simulate line resistance effects. We also removed SiO_x results as the data only modelled programming non-linearity which is less relevant for inference.
2. Performed more realistic line resistance simulations in which weights would be distributed over multiple smaller crossbar arrays. This also meant that the negative effect of line resistance was smaller due to smaller accumulation of currents.
3. Additionally, we explored in more detail the effect of using various reordering schemes to mitigate the effects of line resistance.
4. Although for no other reason rather than just consistency, it is important to mention that the we now explore ANN architectures with one hidden layer, containing 25, 50, 100 and 200 hidden neurons.
5. We renamed the title of our manuscript because we believe that the results summarised in our work are applicable to all resistive memory devices, that nowadays are usually referred to as memristors.

Responses to Individual Comments

Reviewer 1

The authors have addressed my previous questions and significant improved the manuscript, which is essentially acceptable now. A remaining minor concern I have is that Figure 2A in the revised manuscript has been published before in previous publications of some of the co-authors. The authors may want to cite these previous publications in the figure caption to avoid any possible confusion.

We thank the reviewer for noticing this error. In the end, we decided not to use that particular image at all, but make it clear that Ta/HfO₂ data were provided to us in the contributions section.

Reviewer 2

The authors revised the manuscript adding measurements also on array level, which is highly appreciated by this reviewer. Overall the topic is interesting but the paper organization in my opinion lacks of coherence. The major issue are summarized in the following points that were also partially raised during the previous review stage:

First of all, we want to use this opportunity to thank the reviewer, whose thoughtful comments helped us to improve the quality of the manuscript further. We agree with the reviewer's suggestions, and we have now completely redone the paper in light of these comments.

1- I understand that multiple networks work in parallel, but making things work in parallel it's not easy. It is ok to state at line 68 (introduction) that is not a problem, but the the authors should present an estimate of the resource needed later on the main text. Even if the number of weights does not change one has to acquire multiple outputs (it's not possible to have so many ADCs for example), compute the neuron activation and process it to the next neural network layer. At the end the ensemble has to be computed. How this would impact the latency and energy performance?

It is true that if we simply add more networks of the same size, not only does the number of weights increase, but also the number of neurons increases. However, if we constrain ourselves to keeping the same total number of memristors when replacing a large single neural network with a committee of smaller networks, then the number of neurons does not increase significantly. For example, if a single-hidden-layer network with 50 hidden neurons is replaced by a committee of two networks, each with 25 hidden neurons, the total number of hidden neurons effectively does not increase (except for one bias neuron). Furthermore, because all the networks in a committee would share the same inputs, additional input neurons are not necessary, only connections to relevant crossbar word lines, as shown in Figure R1.

In a committee, compared to a large network containing the same total number of synapses, mostly just the number of output neurons increases. But usually it does not increase by much because there are far fewer classes than there are parameters in a network. Figure R2 shows the median accuracy achieved by individual networks and their committees, when controlled for total number of memristors; next to each data point, the total number of neurons is displayed. For a given total number of memristive devices required, we can see that with each additional network in a committee, we are only adding 11 neurons (one bias neuron from the hidden layer and 10 output neurons). Of course, one also has to implement averaging, but if the activation function of the last neuronal layer can be implemented in hardware, so can the averaging by, for example, summing corresponding currents, as shown in Figure R1.

Importantly, the latency is not affected by using committees of smaller networks. *All* members of the committee work in parallel, and the results are only combined at the very last stage

Figure R1: Hypothetical implementation of a committee of two ANNs.

Figure R2: Median accuracy achieved by individual one-hidden-layer memristor-based networks and their committees, when controlled for total number of memristors required. Additionally, next to each data point there is the total number of neurons required for a particular configuration. The networks contained 25, 50, 100 or 200 hidden neurons and were disturbed using faulty devices and D2D variability data from Ta/HfO₂ crossbar.

where we do averaging. Therefore, we believe that energy efficiency will be not be affected by a very modest increase of neurons, and the latency would not be affected because of the parallel nature of computing within the committee. Once again, we thank the reviewer for raising this important issue. We now explain these points in more detail in the main text.

2- Non-linear programming. The authors hasn't still convinced me that non-linear programming is an issue for inference. Programming is a one-time operation for an inference machine, so in principle one could spend a lot of time programming the device with precise program and verify scheme. How the non-linearity simulation were performed? Did any program and verify algorithm were included? In the reviewer opinion, non-linear programming has nothing to do with a precise inference and it is out of the scope of this paper. The authors might present another work were training is taken into account and in that case non-linearity has a huge impact on performances.

We thank the reviewer and fully agree that excluding non-linearity effects makes the main point of the paper easier to follow. Following the suggestion, we decided to completely redo simulations involving programming of conductances. We no longer consider the programming non-linearity and instead consider a verification scheme which would be able to set the devices to the most desirable state by applying multiple SET and RESET pulses. The analysis of Ta/HfO₂ data now includes consideration of faulty devices (stuck in certain state) and device-to-device variability (in a form of different conductance ranges). A much more detailed description is now included in the main text and we believe it presents a realistic picture of how the devices could be programmed.

3- Same thing apply for cycle to cycle (C2C) variability. Why should it be a problem for inference? I have to program the device only one time. In the reviewer opinion also the C2C consideration should be removed from this work a kept for a second work focused on training

Following the reviewer's suggestion, we have now completely removed C2C simulations from the manuscript. C2C is considered in the analysis of Ta/HfO₂ data, but in the main text we explain how it is not relevant for inference.

4- In general the organization of the paper is confusing: Non linear programming is simulated with data from Ta/HfO₂ array and SiO_x device, while RTN is simulated with Ta₂O₅ and a VCMO data and C2C variability with only Ta₂O₅ data. The reviewer thinks that a correct benchmark should be obtain with all the data from all the technology.

To have an high impact and quality paper I suggest to re-organized the paper as follow

1- present the commitment machine

2- measure at array level in the Ta/HfO₂ array the commitment machine accuracy

3- measure at array level in the Ta/HfO₂ array a neural network with the same number of weights as (2) and compare the results

4- measure and characterize RTN on the Ta/HfO₂ array and estimate the parasitic interconnect resistance, measure and characterize RTN in SiO_x, Ta₂O₅, a VCMO devices and simulate the effect of RTN and interconnects for commitment machines and simple neural networks, comparing different technology showing that the technique is technology agnostic.

7- remove completley the effect of non-linear programming and C2C variability

Right now it seems that the authors have used data from different technologies and groups without a coherent characterization procedure that should be the same and extract the same type of data for all the technology to have a fair comparison.

Upon reflection, we agree with the reviewer, and we thank for the very useful directions on how to organise the structure of the paper. We have completely reorganized the paper and, we believe, it now largely follows the structure laid out by the reviewer. Although not all technologies exhibit all the non-idealities, we did combine available experimental data for each of them. Specifically, in the case of Ta/HfO₂ devices, we now combine faulty devices and D2D variability data, as well as line resistance. Ta/HfO₂ devices do not exhibit any apparent RTN and their retention is excellent. Of course, this is a manifestation of one of many trade-offs—memristors with lower resistance exhibit more stable behaviour but are affected more by line resistance. For the Ta₂O₅ and aVMCO devices, we combine their RTN data and use the same interconnect resistance as in Ta/HfO₂ to simulate line resistance effects.

Using these much more detailed descriptions of the technologies, we present the results of committees for networks with 25 hidden neurons. We now have a direct comparison of how much committee machine approach improves the accuracy of different technologies. For further, more general judgement of effectiveness of committees we evaluate their accuracy, while controlling for the total number of devices required. For this, we omit line resistance due to the very long time required to simulate its effects, especially when implementing large neural networks (which are necessary for this comparison). As mentioned before, we completely removed the effect of non-linear programming and C2C variability, which also meant removing SiO_x results due to the fact that it was only used for non-linear programming simulations. We hope that the restructured paper provides a much clearer picture of different non-idealities, technologies and committees.

I understand that this is a lot of work and it looks like a complete new paper, but the authors have in their hands a very powerful tool that if well organized and presented can attract strong interest from the community and high impact. In fact RTN (and other noise issues) and parasitic interconnects resistance drops are huge problems that need to be solved, and I think that commitment machine solution is very promising but should correctly be characterized and presented.

For these reasons, the reviewer thinks that while the work is very interesting it should be reorganized before publication on Nature Communications

Once again, we thank the reviewer for very thoughtful comments that helped us a lot to improve the clarity and quality of the manuscript significantly. We are very pleased to see that the reviewer shares our opinion that this is a powerful tool that in combination with other optimisation techniques (or alone) will help the field to move forward towards viable memristor-based ANN solutions. We hope that the current version of the manuscript is ready

to be published in Nature Communications.

Reviewer 3

The changes that the authors have made have improved the quality of the paper. The results of the network are still poor, however, and I would urge the authors to try another, more relevant, dataset besides MNIST. By trying another dataset you might be able to demonstrate further advantages of your technique.

The paper is very well written and the images are clear.

We thank the reviewer for the comments and for helping us to improve the quality of the manuscript further. We would firstly like to clarify the novelty of our manuscript. We believe that this contribution needs to be seen in the wider context of memristor-based ANN systems. Such systems, compared to CMOS implementations (CPU or GPU systems), promise to bring order of magnitude improvements in power-efficiency. Many papers explore how much energy improvement is possible [1]. The main benefit of memristor-based approaches lies in the fact that these systems integrate memory and computing in one physical location (in-memory computing) and mitigate very costly data transfer. For example, to load data from off-chip memory could be orders of magnitude more energy costly than computations alone [2]. However, the main challenge for memristor analogue devices lies in device imperfections. These negatively impact the performance, such as accuracy explored in this manuscript. Here, we design a system-level solution that can be used in combination with other techniques or alone to mitigate some of these adverse effects. It is true that the MNIST accuracies presented in this manuscript are not as high as those achieved by state-of-the-art architectures implemented in digital computers. However, we think that it is important to consider two contextual points when evaluating our results.

Firstly, the types of networks we explore are relatively simple and small in size. We limit ourselves to fully connected networks that, unlike state-of-the-art digital solutions, do not contain thousands of hidden neurons. This is largely due to the fact that crossbar arrays are most naturally suited to fully connected ANNs and that with the current state of memristive technologies, it is difficult to implement large models on crossbar arrays. Furthermore, it would be impractical to simulate very large neural networks and take the effect of all the non-idealities into consideration at the same time. For example, simulating line resistance effects for even small networks takes us several days when utilising a computer cluster at our university. We thus believe that our results should be viewed in the context of neural networks of at least similar size. Nowadays, fully connected networks are not as often utilised as, for example, CNNs, for visual recognition tasks but reference [3] provides a good summary of MNIST accuracies achieved by simple fully connected networks. It shows that the accuracy is often in the range of 95-98% for fully connected networks containing hundreds of hidden neurons. Now that we removed some of the non-idealities less relevant for inference, all accuracies achieved by our non-ideal memristor-based committee machines lie in that range. For example, Figure R3 shows

the accuracy of memristive ANNs and committees affected by both RTN and line resistance—in committees of 5 networks (each containing only 25 hidden neurons), the median accuracy increases to $\sim 96.6\%$.

Figure R3: Accuracy achieved by individual networks and their committees when RTN data of an aVMCO device are taken into account. Additionally, interconnect resistance of $0.3\ \Omega$ (from Ta/HfO₂ array) was used to include line resistance effects. The maximum whisker length is set to $1.5 \times \text{IQR}$.

Secondly, it is important to evaluate our results in the context of other memristive implementations. Depending on the implementation (in-situ or ex-situ), the technology and the severity of the non-idealities, the reported MNIST accuracies range anywhere from 83% to 95% [4–6] for memristive neural networks. The main challenge of analogue implementations of ANNs is that they currently cannot achieve state-of-the-art accuracies demonstrated by their digital counterparts. The goal of this work was, essentially, to address this problem with a method that could be applied to any memristor-based implementation of neural networks. Our aim with this manuscript is to explore the concept of ensemble averaging in the specific context of memristive neural networks. We do not claim that the accuracy presented in this work is state-of-the-art in the field of neuromorphic engineering. We claim that this method is universal and could be applied to any memristor-based ANN and thus potentially increase its accuracy. We show in the manuscript that the accuracy of a committee depends on the accuracy of individual non-ideal networks which are affected by the non-idealities. Our aim here was not to minimise the non-idealities, but rather to show that committees could be used to deal even with the most severe of them. We believe that if this method is combined with careful optimisation of the physical system, it could yield much higher accuracies.

Additionally, following the suggestion, we decided to test the effectiveness of CM method with a different data set. We decided to use the CIFAR-10 data set, on which we have trained 25 CNNs. These networks had 4 convolutional layers and two fully connected synaptic layers (with a hidden neuronal layer containing 1024 neurons). Because crossbar-based implementations are

most suited to fully connected synaptic layers of neural networks, only the fully connected layers were assumed to be implemented using memristors, while the convolutional layers were assumed to be implemented perfectly using digital computers. That might be a realistic application as fully connected layers usually contain more weights than do convolutional layers.

The results for the accuracies of individual CNNs and their committees are shown in Figure R4. We observe that just like in fully connected networks, the accuracy can significantly increase for non-ideal networks when they are combined into committees. However, we do not present these results in the manuscript as we are focusing on fully-connected networks.

Figure R4: Accuracy achieved by individual CNNs and their committees when faulty devices and D2D variability data of Ta/HfO₂ crossbar are taken into account. The maximum whisker length is set to $1.5 \times \text{IQR}$.

We hope we have addressed the reviewer’s concerns, and we are very excited to share this work with the rest of the community.

References

- [1] Tayfun Gokmen and Yurii Vlasov. “Acceleration of deep neural network training with resistive cross-point devices: Design considerations”. In: *Frontiers in Neuroscience* 10 (2016). doi: [10.3389/fnins.2016.00333](https://doi.org/10.3389/fnins.2016.00333), p. 333.
- [2] Vivienne Sze, Yu-Hsin Chen, Tien-Ju Yang, and Joel S Emer. “Efficient processing of deep neural networks: A tutorial and survey”. In: *Proceedings of the IEEE* 105.12 (2017). doi: [10.1109/JPROC.2017.2761740](https://doi.org/10.1109/JPROC.2017.2761740), pp. 2295–2329.
- [3] Yann LeCun, Léon Bottou, Yoshua Bengio, and Patrick Haffner. “Gradient-based learning applied to document recognition”. In: *Proceedings of the IEEE* 86.11 (1998). doi: [10.1109/5.726791](https://doi.org/10.1109/5.726791), pp. 2278–2324.
- [4] Geoffrey W Burr, Robert M Shelby, Severin Sidler, Carmelo Di Nolfo, et al. “Experimental demonstration and tolerancing of a large-scale neural network (165 000 synapses) using phase-change memory as the synaptic weight element”. In: *IEEE Transactions on Electron Devices* 62.11 (2015). doi: [10.1109/TED.2015.2439635](https://doi.org/10.1109/TED.2015.2439635), pp. 3498–3507.
- [5] Can Li, Daniel Belkin, Yunning Li, Peng Yan, et al. “Efficient and self-adaptive in-situ learning in multilayer memristor neural networks”. In: *Nature communications* 9.1 (2018). doi: [10.1038/s41467-018-04484-2](https://doi.org/10.1038/s41467-018-04484-2), p. 2385.
- [6] Shinhyun Choi, Scott H Tan, Zefan Li, Yunjo Kim, et al. “SiGe epitaxial memory for neuromorphic computing with reproducible high performance based on engineered dislocations”. In: *Nature Materials* 17.4 (2018). doi: [10.1038/s41563-017-0001-5](https://doi.org/10.1038/s41563-017-0001-5), pp. 335–340.

REVIEWER COMMENTS

Reviewer #1 (Remarks to the Author):

The authors have addressed my previous concerns in the revision, which might be accepted now.

Reviewer #2 (Remarks to the Author):

The authors revised the manuscript and with a notable effort, addressed most of the concerns that this reviewer raised. I only have a single major comment. The authors have the access to a crosspoint array of Ta/HfO_x RRAM, thus I suggest to take the opportunity to test the committee machine approach in hardware. I understand that the array size is limited, but with a similar array it is possible to inference a neural network as in reference [15]. In my opinion given the current state of the art of in-memory computing a paper that presents an algorithmic improvement and not a technology (the devices are well known) or circuit (the operation is a matrix-vector-multiplication) should include a array level measurements of such improvement. In this paper case, it would be

1 - a neural network inference without CM

2 - neural network inference with CM and the same number of device of 1

Again, since the authors have a working array available on which a neural network inference and training had already been presented, I don't see any obstacle of doing that.

Some minor comments:

1 - I would be interesting to have a more quantitative comparison of the two approaches of figure 1 B-C

2 - Figure 2 B-F are beautiful but it took me a while to read it and understand them. I suggest to use a less complicated G vs pulse number cartesian plot.

3 - line 249 misspelled eight

Major Changes to the Manuscript

We thank for additional comments and suggestions. By taking them into account, in addition to some other changes, we hope to have improved the quality of the manuscript even further. The most important changes to the manuscript are listed below.

1. We have redone all the simulations involving line resistance. Previous versions of the manuscript used a scheme in which two memristors implementing the same weight would not necessarily be placed next to each other. Although we mentioned that this is a naive scheme in the previous version of the manuscript, we had underestimated how much it decreases the accuracy. We now use a more conventional scheme in which the two memristors implementing the same weight are placed in neighbouring bit lines. The accuracy achieved is even higher now and all the qualitative results still hold.
2. Because the new scheme achieves much better accuracy, we decided to no longer use or explain intensity-aware mapping in the main text; after all, the main idea of the manuscript is committee machines. However, we explain this scheme in the supplementary information because it can significantly improve inference accuracy.
3. We decided to no longer explore random reordering of word and bit lines to increase variability of neural networks of a committee when they are affected by line resistance. Firstly, it is extremely difficult to evaluate the effectiveness of this method while controlling for every other variable that could affect the accuracy. Secondly, even when we manage to control for other variables to a satisfying extent, we find that the effect size, at least with MNIST data set and the small size of our networks, is minimal. Due to lack of strong evidence, we do not want to claim that this method improves variability of networks affected by line resistance at this time.
4. We added Supplementary Figure S8 to show that using different digital networks in a committee, significantly increases its accuracy.
5. We added Supplementary Figure S9 which shows pulsing information from Figures 2B-F in Cartesian plots.

Responses to Individual Comments

Reviewer 2

The authors revised the manuscript and with a notable effort, addressed most of the concerns that this reviewer raised. I only have a single major comment. The authors have the access to a crosspoint array of Ta/HfO_x RRAM, thus I suggest to take the opportunity to test the committee machine approach in hardware. I understand that the array size is limited, but with a similar array it is possible to inference a neural network as in reference [15]. In my opinion given the current state of the art of in-memory computing a paper that presents an algorithmic improvement and not a technology (the devices are well known) or circuit (the operation is a matrix-vector-multiplication) should include a array level measurements of such improvement. In this paper case, it would be

1 - a neural network inference without CM

2 - neural network inference with CM and the same number of device of 1

Again, since the authors have a working array available on which a neural network inference and training had already been presented, I don't see any obstacle of doing that.

Some minor comments:

1 - I would be interesting to have a more quantitative comparison of the two approaches of figure 1 B-C

2 - Figure 2 B-F are beautiful but it took me a while to read it and understand them. I suggest to use a less complicated G vs pulse number cartesian plot.

We thank the reviewer for carefully reviewing our revised manuscript. We have now made additional changes with regard to the minor comments as suggested.

Firstly, we agree that demonstrating differences in accuracy improvement when using same and different digital networks in a committee would be useful. We have thus included such a comparison (see Figure R1) in the supplementary information. It shows that using different digital networks allows us to achieve higher accuracy. In the case of Figure R1, we see that committees of size 5 utilising different digital networks achieve $\sim 1.1\%$ higher median accuracy than committees using the same digital networks.

We also thank the reviewer for the comment regarding the plots of pulsing data. We have now included a Cartesian version (shown here in Figure R2) of these plots in the supplementary information. However, we left the polar plots in the main text. Although we agree they are less conventional, we believe they fit the cyclical nature of the pulsing data better. Additionally, they draw attention away from any linearity in the pulsing behaviour which is not important for the purposes of ex-situ training that we focus on in this manuscript.

Figure R1: Accuracy achieved by individual networks and their committees when faulty devices and D2D variability data of Ta/HfO₂ crossbar are taken into account. **A)** Using identical digital networks when implementing committees of memristive neural networks. **B)** Using different digital networks when implementing committees of memristive neural networks. The maximum whisker length in both subfigures is set to $1.5 \times \text{IQR}$. The accuracy of individual disturbed non-ideal memristive networks in the two subfigures is not *identical* only because the data were produced using two different simulations. However, it is clear that using different digital networks results in higher accuracy of the committees.

Figure R2: Pulsing data from Figures 2B-F represented in Cartesian form. 11 SET cycles are depicted in blue, while 11 RESET cycles are depicted in orange. **A)** Equivalent of Figure 2B. **B)** Equivalent of Figure 2C. **C)** Equivalent of Figure 2D. **D)** Equivalent of Figure 2E. **E)** Equivalent of Figure 2F.

Now we would like to address the important question of a hardware demonstration.

Although such demonstration of the committee machines would further increase the impact, at the moment, our platform is not suitable for his task.

To appropriately test the CM design fully in hardware we would need to connect a large number of arrays together, as well as implement hardware neurons. However, currently, we connect one probe station to one array on a chip, and our previous multilayer ANNs were implemented by partitioning a 128×64 array into different parts. This approach was suitable only for very small neural networks (e.g. 8×8 inputs in comparison to MNIST inputs of size 28×28 , that we use in this paper) [1] or to store kernels of convolutional neural networks (as in [2]). The reference [3] (referred to as [15] in the reviewer's comments) does not include inference or training of

ANNs, but image compression and convolutional filtering for which a single crossbar is enough. Nevertheless, even for this one crossbar, a designated measurement system had to be made (see Figure R3). Connecting multiple arrays to accommodate committees of multi-layer ANNs is not possible on our experimental platform as it would require a designated measurement system for each separate array; they are currently not available in our lab.

[REDACTED]

Supplementary Figure 2 from Li, C. et al. Analogue signal and image processing with large memristor crossbars. *Nat. Electron.* **1**, 52–59 (2018).

Taken from the supplementary information of [3].

The partitioning of the crossbar is not practical as the CM approach requires parallel measurements, and we would still need to have multiple measurement systems working in parallel. Moreover, even with only a few members of the committee and very small neural networks, a single 128×64 array would not be suitable. The crossbar would need to be partitioned both across members of the committee, as well as across different layers of individual ANNs. This approach would lead to a very small number of available devices per ANN layer.

Therefore, we have used the hybrid approach (using experimental array-level data from 128×64 crossbar combined with simulations). This was the addition after our 1st revision, which the reviewer kindly acknowledged (“The authors revised the manuscript adding measurements also on array level, which is highly appreciated by this reviewer.”).

We believe that by fully accounting for device non-idealities (faulty devices, variability and RTN), as well as system non-idealities (line resistance)—all obtained by measuring multiple different RRAM technologies—we take into account all major adverse effects for inference accuracy. As correctly stated by the reviewer, the main novelty of this work is the idea of CM, not the demonstration of a mature technology (device technology, or fully integrated memristor system). We do not rely on any device modelling, but rather conduct measurements on real crossbar arrays and appropriately analyse data for the CM testing.

We were very encouraged by the reviewer’s comment after the second revision:

I understand that this is a lot of work and it looks like a complete new paper, but the authors have in their hands a very powerful tool that if well organised and presented can attract strong interest from the community and high impact. In fact RTN (and other noise issues) and parasitic interconnects resistance drops are huge problems that need to be solved, and I think that commitment machine solution is very promising but should correctly be characterised and presented.

We have taken on board this useful remark by completely restructuring the manuscript accordingly and by incorporating all array-level experimental data over the last two revisions. We believe strongly that the idea of the CM will be very impactful, as it deals with the most significant problems of memristor technologies. The benefit of our method is that it can be used with any other improvement strategy (material optimisation, defect-aware training, etc.), and it is truly technology- and non-ideality-agnostic. This is a unique advantage of our approach and we believe it is widely applicable (as demonstrated by using three different technologies and accounting for all the main device non-idealities). We do not feel that a full hardware demonstration would add anything significant to the core of our work at this stage; the hybrid approach we have taken is a clear demonstration of the potential of the CM approach. Importantly, as it is technology-agnostic, a full hardware demonstration on a specific platform, while a nice thing to have, would require such significant investment and delay that the timeliness and utility of our approach would be compromised.

We sincerely thank the reviewer for the effort and all the valuable comments during the last four rounds of revisions. We hope that our manuscript is now ready to be published in Nature Communications.

References

- [1] Can Li, Daniel Belkin, Yunning Li, Peng Yan, et al. “Efficient and self-adaptive in-situ learning in multilayer memristor neural networks”. In: *Nature communications* 9.1 (2018). doi: [10.1038/s41467-018-04484-2](https://doi.org/10.1038/s41467-018-04484-2), p. 2385.
- [2] Zhongrui Wang, Can Li, Peng Lin, Mingyi Rao, et al. “In situ training of feed-forward and recurrent convolutional memristor networks”. In: *Nature Machine Intelligence* 1.9 (2019). doi: [10.1038/s42256-019-0089-1](https://doi.org/10.1038/s42256-019-0089-1), pp. 434–442.
- [3] Can Li, Miao Hu, Yunning Li, Hao Jiang, et al. “Analogue signal and image processing with large memristor crossbars”. In: *Nature Electronics* 1.1 (2018). doi: [10.1038/s41928-017-0002-z](https://doi.org/10.1038/s41928-017-0002-z), pp. 52–59.